# Insights into the κ/ι-carrageenan metabolism pathway of some marine *Pseudoalteromonas* species

Andrew G. Hettle[1], Joanne K. Hobbs[1], Benjamin Pluvinage[1], Chelsea Vickers[1,4], Kento T. Abe[1,5], Orly Salama-Alber[1], Bailey E. McGuire[1], Jan-Hendrik Hehemann[1,6], Joseph P.M. Hui[2], Fabrice Berrue[2], Arjun Banskota[2], Junzeng Zhang[2], Eric M. Bottos[3], Jonathan Van Hamme[3] & Alisdair B. Boraston[1]*

*Pseudoalteromonas* is a globally distributed marine-associated genus that can be found in a broad range of aquatic environments, including in association with macroalgal surfaces where they may take advantage of these rich sources of polysaccharides. The metabolic systems that confer the ability to metabolize this abundant form of photosynthetically fixed carbon, however, are not yet fully understood. Through genomics, transcriptomics, microbiology, and specific structure-function studies of pathway components we address the capacity of newly isolated marine pseudoalteromonads to metabolize the red algal galactan carrageenan. The results reveal that the κ/ι-carrageenan specific polysaccharide utilization locus (CarPUL) enables isolates possessing this locus the ability to grow on this substrate. Biochemical and structural analysis of the enzymatic components of the CarPUL promoted the development of a detailed model of the κ/ι-carrageenan metabolic pathway deployed by pseudoalteromonads, thus furthering our understanding of how these microbes have adapted to a unique environmental niche.

[1] Department of Biochemistry and Microbiology, University of Victoria, PO Box 1700 STN CSCVictoria, British Columbia V8W 2Y2, Canada. [2] Aquatic and Crop Resource Development Research Centre, National Research Council of Canada, 1411 Oxford Street, Halifax, NS B3H 3Z1, Canada. [3] Department of Biological Sciences, Thompson Rivers University, 805 TRU Way, Kamloops, British Columbia V2C 0C8, Canada. [4]Present address: School of Biological Sciences, Victoria University, PO Box 600 , Wellington 6012, New Zealand. [5]Present address: Lunenfeld-Tanenbaum Research Institute, Sinai Health System, and Department of Molecular Genetics, University of Toronto, 600 University Ave, Rm 992, Toronto, ON M5G1X5, Canada. [6]Present address: Marum and Max Planck Institute for Marine Microbiology, Celsiusstraße 1, 28359 Bremen, Germany. *email: boraston@uvic.ca

The marine environment plays host to approximately one-half of all global primary production[1]; however, the contribution to this by macroalgae is predominantly confined to coastal ecosystems[2]. Macroalgal surfaces offer a unique habitat for surface-associated microorganisms such as fungi[3,4] and yeast[5], as well as both Gram-positive[6] and Gram-negative bacteria[7]. Surface-associated bacteria can colonize in the range of $10^6$–$10^9$ bacteria/cm$^2$ of algal surface area[7–9] where the bacterial community makeup is influenced by the polysaccharide composition of the host algae[10]. The polysaccharide composition aids in selecting for genetic profiles of heterotrophic marine bacteria that contain the complement of functional genes required for polysaccharide catabolism, which are often localized within PULs (Polysaccharide Utilization Loci)[11]. This cycle of polysaccharide synthesis by marine algae and polysaccharide catabolism by heterotrophic microbes comprises one of the largest and fastest biotransformations on earth, making it a key component in the global carbon cycle[12,13].

Roughly 50% of macroalgal dry weight comes from cell wall polysaccharides[14,15], which are a large component of the photo-synthetically fixed carbon found in the marine environment[16]. The polysaccharides in red algae (Rhodophyta) are primarily mixed galactan agars such as agarose and carrageenan. Carrageenans are high molecular weight polymers having a linear backbone of repeating D-galactopyranose units (G) linked together by alternating α-1,3 and β-1,4 linkages. The β-linked galactopyranose can also exist as a unique bicyclic residue, 3,6-anhydro-D-galactose[17] (DA), giving rise to the disaccharide building block β-neocarrabiose (β-NC2), which is the base structure of beta(β)-carrageenan. The carrageenan class is dependent on the presence or absence of these DA units and on the degree of free hydroxyl group sulfation at C2, C4, and C6 (2S, 4S, 6S, respectively) giving rise to a large number of possible carrageenan families[18–20] with the most common being kappa(κ)-carrageenan and iota(ι)-carrageenan. These two families both bear 4-O-sulfation on the G units (G4S), while ι-carrageenan also contains 2-O-sulfation on the DA units (DA2S). The complexity of carrageenan increases with the potential presence of hybrid carrageenan polymers that consist of neocarrabiose units from different families linked together, along with other potential decorations, like methyl-esters and pyruvate groups[20,21].

The study of carrageenan metabolism has primarily focused on enzymes involved in the initial steps of carrageenan depolymerization: endo-acting κ-carrageenases and ι-carrageenases from glycoside hydrolase (GH) families 16 and 82, respectively[22–25], as well as endo-acting carrageenan sulfatases[26–29]. Many of these characterized carrageenan-specific enzymes are from species of Pseudoalteromonas, which is a marine-associated genus of heterotrophic gamma-proteobacteria found in a broad range of marine environments[30,31]. The historic relationship of Pseudoalteromonas species with carrageenan degradation has made this a model genus for understanding microbial carrageenan metabolism. However, a recent leap with respect to carrageenan degradation in general was made through the discovery and characterization of a carrageenan-specific PUL (CarPUL) in Zobellia galactanivorans[32]. The carrageenan-active enzymes deployed by this microbe include endo-acting carrageenases (GH16 and GH82), exo-acting carrageenan-specific GHs (GH2, GH127, and GH129) and carrageenan-specific sulfatases (S1_7, S1_17, and S1_19) that confer upon the microbe the ability to depolymerize and metabolize κ- and ι-carrageenan. Subsequently, Gobet et al., using a bioinformatics approach based on the genome sequence of Pseudoalteromonas carrageenovora 9$^T$, proposed a carrageenan metabolism pathway for marine pseudoalteromonads[33]. This proposed pathway, which displays predicted functional features distinct from those of the Z. galactanivorans

pathway (e.g., different GH families and different sulfatase subfamilies), remains to have many steps functionally verified. Most notably, however, P. carrageenovora 9$^T$ was found to be unable to use κ- or ι-carrageenan as an energy source[33], raising the question of the general metabolic relevance of the CarPUL in pseudoalteromonads.

In this study, we report the comparison of five Pseudoalteromonas species isolated from the surface of red macroalgae found in the intertidal zone of south Vancouver Island, Canada. The Pseudoalteromonas CarPUL was found in four of the isolates and these were the only ones to display growth on κ- and ι-carrageenan. Furthermore, transcriptomics revealed upregulation of the CarPUL during growth on ι-carrageenan, while biochemical analysis of eight CarPUL-encoded enzymes revealed carrageenan-specific activity, thus associating the CarPUL with the ability of these pseudoalteromonads to liberate energy from carrageenan. Detailed analysis of the enzyme activities encoded by the CarPUL, combined with interpretation of growth phenotypes, have allowed us to propose a refined and detailed model of κ- and ι-carrageenan processing by some Pseudoalteromonas species.

## Results

**A carrageenan PUL conserved amongst some Pseudoalteromonas species.** Five bacterial isolates (referred to herein for brevity as PS2, PS42, PS47, FUC4, and U2A) were cultured from the surface of Fucus sp. obtained from a coastal marine environment (Willows Beach, Victoria, BC, Canada) and the genomes of these isolates were sequenced (Supplementary Table 1). Analysis of 16S rRNA gene sequences identified these isolates as belonging to the genus Pseudoalteromonas (Supplementary Table 1). Whole-genome average nucleotide identities (ANI) identified PS2, PS42, and PS47 as strains of Pseudoalteromonas fuliginea (Supplementary Table 1). Comparison of the PS2 and PS47 genome sequences resulted in an ANI value of 99.99% over 4.4 Mbp indicating that they are likely independent isolates of the same P. fuliginea strain, though here we treat them as independent strains. Whole-genome ANI analysis revealed U2A to be a strain of Pseudoalteromonas distincta. FUC4 was most similar to P. distincta; however, an ANI value of <95% made a firm determination of species unclear based on currently available Pseudoalteromonas species genome sequences.

A survey of the genomes revealed PS2, PS47, FUC4, and U2A to have loci very similar to the recently identified CarPUL from P. carrageenovora 9$^T$ (Fig. 1 and Supplementary Table 2)[33]. The CarPULs only differ by the presence or lack of a limited number of genes, while the proteins encoded by the conserved genes share no <70% amino sequence identity and typically >90–95% amino acid sequence identity (Supplementary Table 2). PS42 lacks this CarPUL thereby representing a naturally occurring strain of P. fuliginea that is deficient of any obvious genetic potential to metabolize carrageenan [see also Supplementary Notes (section 1)].

**A complex growth phenotype on κ/ι-carrageenan.** Consistent with its lack of a CarPUL, PS42 failed to display any growth in liquid culture containing ι-carrageenan or carrageenan oligosaccharides (Fig. 2a). The other four isolates failed to grow when ι-carrageenan, ι-neocarratetraose (ι-NC4), κ-neocarratetraose (κ-NC4), or κ-neocarraoctaose (κ-NC8) (all carbohydrates were tested at 0.4%) were provided as a sole carbon source (Fig. 2a and Supplementary Fig. 1). However, growth on ι-carrageenan was observed for PS47, PS2, U2A, and FUC4 when the cultures were supplemented with an exogenous GH16 carrageenase from Bacteroides ovatus (BovGH16, Fig. 2a), which is active on both κ- and ι-carrageenan (Supplementary Fig. 2). The four isolates

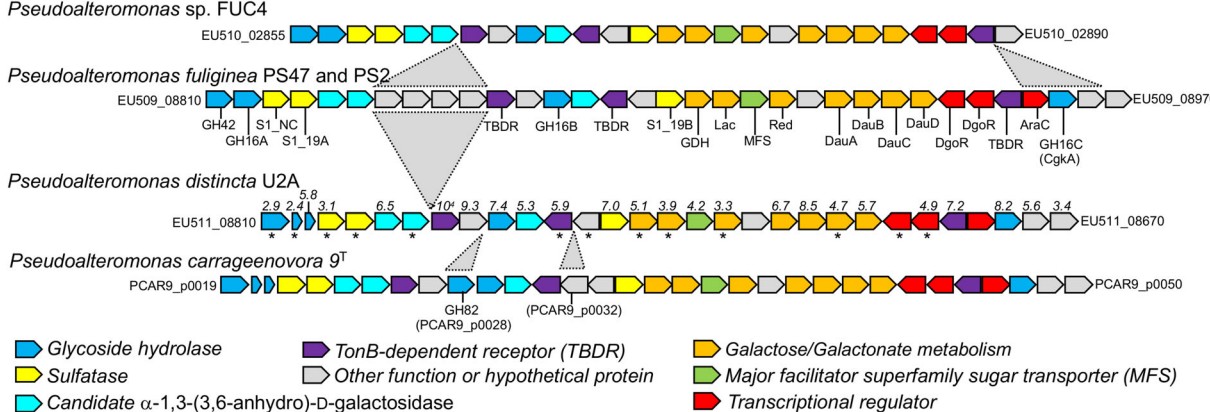

**Fig. 1 Schematic of the carrageenan PULs in newly isolated *Pseudoalteromonas* sp. compared with *Pseudoalteromonas carrageenovora* 9<sup>T</sup>.** Each gene is represented by an arrow, which are color coded according to putative function. Gray triangles indicate inserted regions. All other genes are otherwise orthologous. Numbers above the genes in the *Pseudoalteromonas distincta* U2A CarPUL indicate the Differential Expression $Log_2$ Ratio of transcript levels detected when grown on ι-carrageenan vs. galactose as determined by RNA sequencing experiments (values are only shown for genes with |$Log_2$ Ratios| > 1.5 and *p*-values < 0.05). Asterisks (*) below the genes indicate those with significant transcripts (>10 transcripts per kilobase million) when grown on galactose. Locus tags for the first and last genes are provided. See also Supplementary Table 2 for more details of the gene annotations.

possessing the CarPUL also demonstrated growth when provided with ι-carrageenan as a carbon source if the cultures were supplemented with 0.04% κ-NC4 or κ-NC8, but did not grow when supplemented with ι-NC4 (Fig. 2a and Supplementary Fig. 1).

All of the isolates (pre-grown in a galactose containing liquid medium) grew when spotted on a medium comprising gelled κ-carrageenan if it was supplemented with galactose (Fig. 2b). When galactose was omitted from the gelled κ-carrageenan slow growth was observed with visible colonies appearing over the course of ~7 days for all strains except PS42 (Fig. 2c). More robust growth on the gelled κ-carrageenan was observed within a few days for all isolates, except PS42, when the isolates were first grown in liquid medium containing ι-carrageenan supplemented with 0.04% κ-NC4 (galactose was necessarily added to the pre-growth medium for PS42) (Fig. 2d).

**Growth on carrageenan upregulates the CarPUL.** U2A was the easiest strain to handle and grow on ι-carrageenan in liquid culture (with exogenous addition of BovGH16), thus, we chose this as model for transcriptomic analysis of CarPUL expression by RNASeq (Supplementary Data 1). A control culture grown on galactose showed 13 of 30 genes in the U2A CarPUL (the fragmented *gh16A* gene was counted as two genes) to have significant transcript counts [criteria of >10 transcripts per kilobase million (TPM)] (Fig. 1). This may result from some capacity of the monosaccharide to regulate components the PUL or it reveals the constitutive production of some pathway components. In comparison to CarPUL gene expression in the presence of galactose, 24 of 30 genes showed significantly higher levels of transcript production in the presence of ι-carrageenan (Differential Expression $Log_2$ Ratio of >1.5 and *P*-value < 0.05), indicating the response of the CarPUL to the presence of carrageenan (Fig. 1). Notably, with a TPM value of >1500 when grown on galactose the transcript encoding the U2A orthologue of S1_19A was one of the most abundant transcripts found under either growth condition with no statistical difference in expression between growth on galactose vs. growth on ι-carrageenan (Supplementary Data 1).

The cell-free culture supernatants (CS) harvested from cultures grown in liquid medium containing ι-carrageenan supplemented with 0.04% κ-NC4 showed that the CS from PS2, PS47, U2A, and FUC4 incubated with κ-NC8 as a substrate clearly depleted the κ-NC8 and produced a series of smaller oligosaccharides (Fig. 2e and Supplementary Fig. 3a). This supports the concept that gene

expression is accompanied by the production of carrageenan-active proteins. CS from PS42, which was grown in liquid medium containing ι-carrageenan supplemented with 0.04% κ-NC4 and galactose, appeared unable to depolymerize κ-NC8 and only residual κ-NC4 from the growth medium was detected. We further tested the total cell fraction (TCF) and the CS of PS47 on both ι-carrageenan and κ-carrageenan. The CS had no obvious activity on ι-carrageenan but produced a distinct banding pattern when incubated with κ-carrageenan, indicating depolymerization of κ-carrageenan (Fig. 2f and Supplementary Fig. 3b). The TCF was capable of depolymerizing both types of carrageenan.

**Family 16 glycoside hydrolases from PS47 depolymerize carrageenan.** The CarPUL of PS47 encodes three putative family 16 GHs, which we refer to as GH16A (locus tag EU509_08815), GH16B (EU509_08870), and GH16C (EU509_08960). GH16A and GH16C have significant amino acid sequence identity (>79%) with the well characterized κ-carrageenase CgkA (PCAR9_p0048) from *P. carrageenovora* 9<sup>T,22,34,35</sup> while GH16B shows ~78% amino acid sequence identity with a *Paraglaciecola hydrolytica* S66<sup>T</sup> GH16 enzyme that is classified as a β-carrageenan-specific endo-hydrolase[36] [see Supplementary Notes (section 2) and Supplementary Fig. 4]. GH16A and GH16C displayed activity on κ-carrageenan, producing products consistent with κ-neocarrageenan oligosaccharides with even numbers of sugar residues (Fig. 3a and Supplementary Fig. 5a). These two enzymes had minimal activity when incubated with ι-carrageenan, producing small amounts of product with mobilities similar to that of κ-neocarrageenan oligosaccharides (Fig. 3b and Supplementary Fig. 5b). This suggests activity on a small amount of contaminating κ-carrageenan in the ι-carrageenan, which is consistent with the NMR analysis of this preparation of ι-carrageenan[28].

GH16B had no activity on either κ- or ι-carrageenan (Fig. 3a, b and Supplementary Fig. 5a and b). Given the previously reported capacity of S1_19A (locus tag EU509_08825) to desulfate the G4S residues of κ- and ι-carrageenan[28], which would produce β- and α-carrageenan (and/or hybrids), respectively, we tested its activity in conjunction with the GH16 enzymes. S1_19A pretreated κ-carrageenan resulted in a similar pattern of products when incubated with GH16A and GH16C as did κ-carrageenan when not treated with sulfatase; however, cotreatment with S1_19A resulted in mobility shifts of the products, possibly due to some post-depolymerization desulfation of the oligosaccharides (Fig. 3c

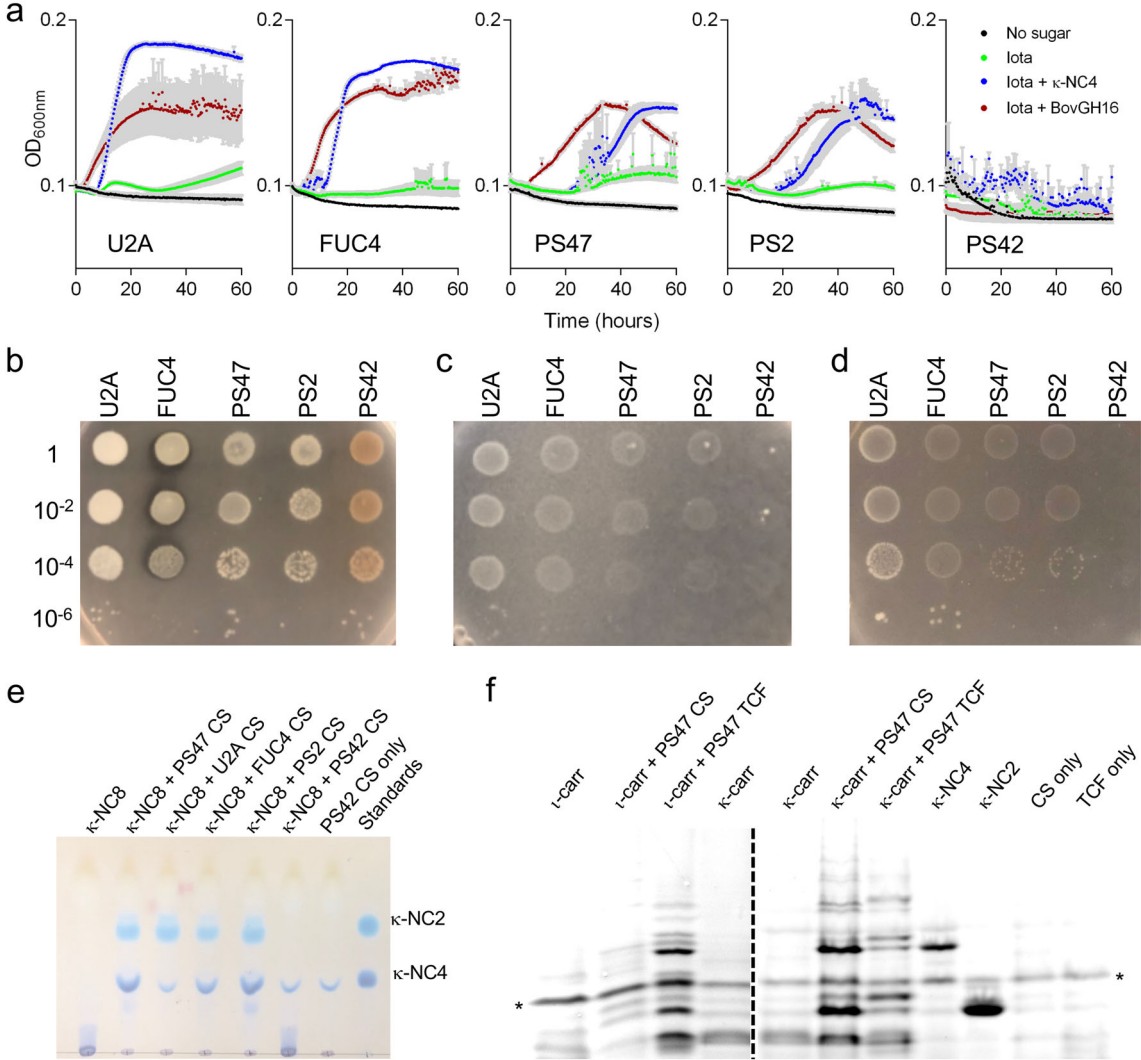

**Fig. 2 Growth and carrageenan degradation properties of *Pseudoalteromonas* isolates. a** Growth of the *Pseudoalteromonas* isolates in liquid minimal marine media (MMM) with no carbon source (black), MMM with 0.4% ι-carrageenan (green), MMM with 0.4% ι-carrageenan and 0.04% κ-NC4 (blue), and MMM with 0.4% ι-carrageenan and BovGH16 (red). Gray bars indicate the error from $n = 4$ independent experiments. **b–d** Growth of the *Pseudoalteromonas* isolates on MMM solidified with 1% κ-carrageenan. In panels (**b**) and (**d**), prior to spotting on the plates, the bacteria were pre-grown in liquid medium comprising MMM, 0.4% ι-carrageenan and 0.04% κ-NC4. In panel (**b**), the solid medium was also supplemented with 1% galactose. For these plates, the pre-growth media was also supplemented with 0.5% galactose to grow PS42. In panel (**c**), the bacteria were pre-grown in liquid medium comprising MMM and 0.5% galactose. The numbers to the left of the panels indicates the fold dilution of the pre-culture that was used to spot the solid medium. **e** Thin layer chromatography (TLC) analysis of κ-NC8 incubated with cell-free culture supernatants (CS) taken from the *Pseudoalteromonas* isolates after growth on MMM supplemented with 0.4% ι-carrageenan and 0.04% κ-NC4 (medium also contained 0.5% galactose for PS42). **f** Fluorophore-assisted carbohydrate electrophoresis (FACE) analysis of ι-carrageenan (ι-carr) and κ-carrageenan (κ-carr) incubated with CS or total cell fraction (TCF) from PS47 after growth on MMM supplemented with 0.4% ι-carrageenan and 0.04% κ-NC4. The asterisks (*) indicate the band corresponding to excess ANTS fluorophore.

and Supplementary Fig. 5c). GH16A and GH16C remained inactive on ι-carrageenan pre- or co-treated with S1_19A (Fig. 3d and Supplementary Fig. 5d). When κ-carrageenan was pretreated with S1_19A before incubation with GH16B, or when GH16B was co-incubated with the sulfatase, GH16B displayed the capacity to depolymerize the resulting carrageenan structure, indicating activity on β-carrageenan or a β/κ-carrageenan hybrid (Fig. 3c and Supplementary Fig. 5c). Treatment of ι-carrageenan with S1_19A also rendered this polysaccharide a substrate for GH16B, indicating the ability of GH16B to hydrolyze α-carrageenan or a α/ι-carrageenan hybrid (Fig. 3d and Supplementary Fig. 5d), which is consistent with the depolymerization observed when incubating ι-carrageenan with PS47 TCF (Fig. 2f and Supplementary Fig. 3b). Unlike with GH16A and GH16C,

the pretreatment of ι-carrageenan with S1_19A followed by GH16B digestion, or cotreatment with both enzymes at the same time, resulted in the same pattern of GH16B products. Because GH16B activity depends on S1_19A activity, we suggest that the former observation indicates that the products of GH16B lack G4S residues that are susceptible to S1_19A activity, and may even entirely lack G4S residues.

**S1_19B from PS47 is an exo-G4S κ-carrageenan sulfatase.** S1_19B (locus tag EU509_08890) is a putative S1 sulfatase falling into subfamily 19 of the sulfatase classification[37]. The potential sulfatase activity of recombinant S1_19B was screened against κ-neocarrabiose (κ-NC2), κ-NC4, ι-neocarrabiose (ι-NC2) and ι-NC4 by thin layer chromatography (TLC), revealing activity on

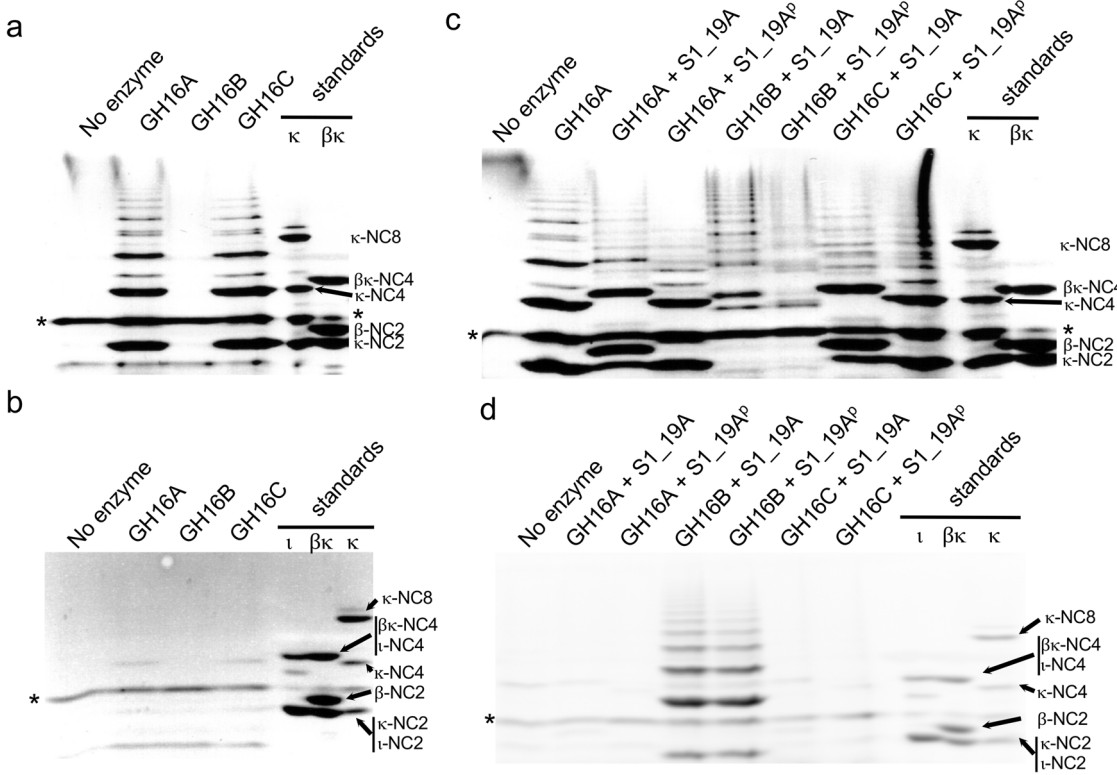

**Fig. 3 The carrageenan degradation properties of recombinant GH16 enzymes from PS47.** FACE analysis of the products of κ-carrageenan (**a**) and ι-carrageenan (**b**) degradation. Panels (**c**) and (**d**) show the FACE analysis of κ-carrageenan (**c**) and ι-carrageenan (**d**) degradation when the GH16 enzymes are used in conjunction with the S1_19A endo-4S-κ/ι-carrageenan sulfatase. The superscript "p" ($^p$) in the sample label indicates that the carrageenan was pretreated overnight with the sulfatase, followed by heat inactivation of the sulfatase then digestion with the GH16. The asterisks (*) indicates the band corresponding to excess ANTS fluorophore.

κ-NC2 and κ-NC4 (Fig. 4a and Supplementary Fig. 6a); however, S1_19B did not appear to have any activity on ι-NC2 or ι-NC4. The $K_M$ and apparent $V_{max}$ values for S1_19B against κ-NC2 were $292.2 \pm 14.0\,\mu M$ and $0.7 \pm 0.1$ nmoles s$^{-1}$, respectively (Supplementary Fig. 7a).

Native S1_19B crystallized as a dimer, a quaternary structure that was confirmed to be present in solution, with the monomeric structure revealing the expected domain architecture typical of S1 sulfatases and containing a coordinated calcium ion in the active site pocket [Supplementary Notes (section 3) and Supplementary Fig. 7b–d]. Subsequently, we determined the structure of S1_19B in complex with intact κ-NC2 substrate. This was enabled by the use of a mutant that was inactivated by a serine substitution of residue C77, which would otherwise be matured to the catalytic formylglycine (FGly) residue. Each of the six S1_19B C77S monomers in the asymmetric unit contained a single intact κ-NC2 molecule with continuous density occupying the active site pocket and spanning the catalytic machinery (Fig. 4b and Supplementary Fig. 7e). The 4-sulfate of the G4S unit in the 0 subsite (sulfatase subsite nomenclature adopted from Hettle et al.[28]) was coordinated in the S subsite by the amino acid sidechains conserved in S1 sulfatases (Fig. 4c). The serine residue replacing the proto-catalytic cysteine was positioned ~3.5 Å below the targeted sulfate ester mimicking the position of what would be the catalytic FGly residue. H233 was positioned ~3.1 Å from the scissile bond and oriented to protonate the ester oxygen in its role as the catalytic acid.

The non-reducing end DA residue in the −1 subsite interacts with the hydroxyl groups of carbons C2 and C4 and with the oxygen of the 3,6-anhydrous bridge through a direct and water-mediated hydrogen bond network. F103, Y455, and Y458 create an "aromatic shelf" aiding in the positioning of the DA unit (Fig. 4d). Notably, this mode of recognition necessitates binding the non-reducing end of the oligosaccharide, not internal chain binding, indicating a specific exo-mode of substrate recognition (Fig. 4b, d).

Though S1_19B was inactive on ι-NC4, the structure of this enzyme in complex with ι-NC4 revealed the ability of the enzyme to bind this oligosaccharide, and accommodation of the 2-sulfate on the non-reducing end residue in the −1 subsite (Supplementary Fig. 4f). However, the presence of this sulfate group appears to cause a shift in the placement of the oligosaccharide in the active site, resulting in the disengagement of the 4-sulfate on the residue in the 0 subsite from the catalytic machinery and formation of an inactive complex (Supplementary Fig. 7f–h). Additional interactions between ι-NC4 and the enzyme suggest the possibility of +1 and +2 subsites (Supplementary Fig. 7g, h). Overall, the data support assignment of S1_19B as an exo-G4S κ-carrageenan sulfatase that is specific for the non-reducing end of κ-carrageenan oligosaccharides.

**Structural analyses of S1_NC from PS47.** As with S1_19B, we screened the potential activity of S1_NC (locus tag EU509_08820), a putative sulfatase in the non-classified (NC) group of S1 sulfatases[37], against a panel of neocarrageenan oligosaccharides, but failed to detect any activity. Initial X-ray crystallographic analysis of S1_NC suggested this was due to aberrant maturation of the proto-catalytic cysteine [Supplementary Notes (section 4) and Supplementary Figs. 8, 9]. Despite extensive efforts, we were unable to circumvent generation of this inappropriately matured form of the protein. Therefore, to provide insight to the specificity of this enzyme we generated C84A

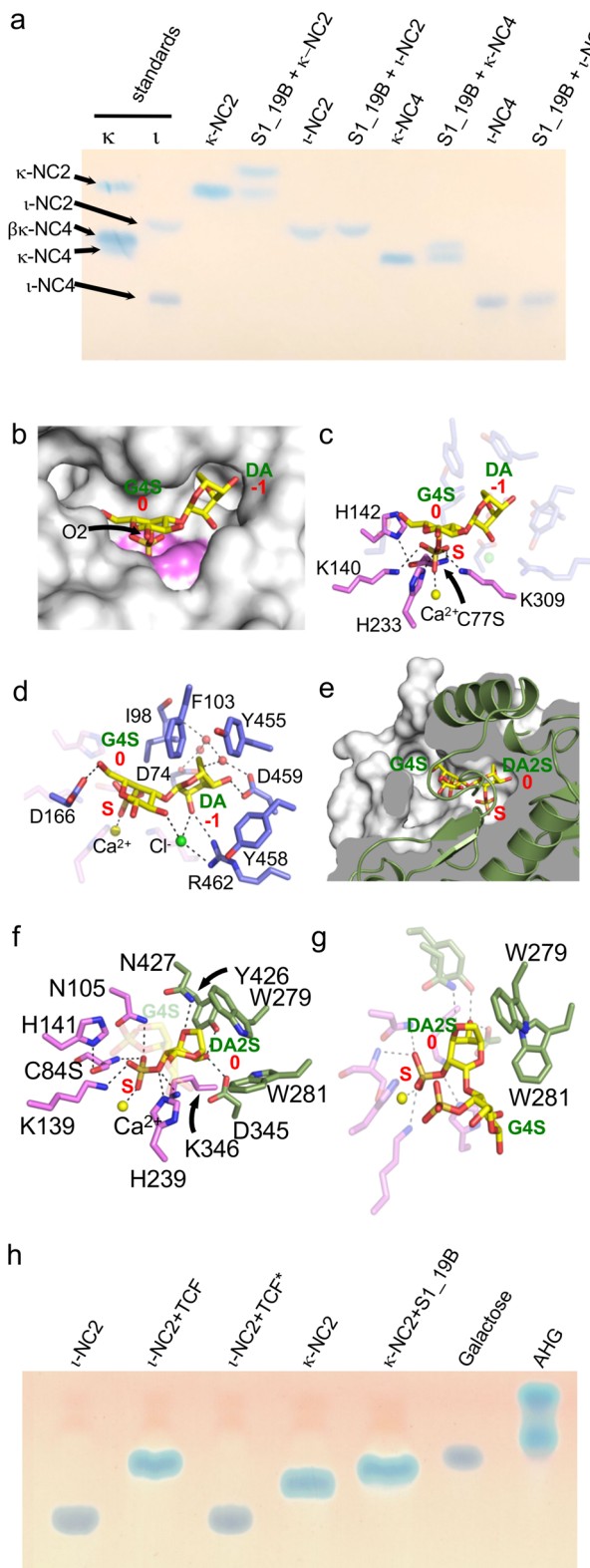

**Fig. 4 Activity and structural features of PS47 sulfatases. a** TLC analysis of S1_19B activity. **b–d** Structural analysis of S1_19B with (**b**) showing the active site pocket of the S1_19B C77S mutant in complex with κ-NC2 as a solvent accessible surface (gray) and the area comprising the S-subsite colored in violet. The bound κ-NC2 is shown as yellow sticks. **c** The interactions of the sulfate ester of κ-NC2 with the sulfate binding S-subsite. **d** The specific interactions of κ-NC2 with the active site pocket 0 and −1 subsites. **e–g** Structural analysis of S1_NC with (**e**) showing the cutaway of S1_NC C84S in complex with ι-NC2, which reveals the tunnel like nature of the active site. S1_NC is shown as a green cartoon and the solvent accessible surface is shown in white. Gray represents the interior of the enzyme and ι-NC2 as shown as yellow sticks. **f** The interactions of the targeted sulfate ester of ι-NC2 with the sulfate binding S-subsite and the DA2S unit with the residues of the 0 subsite. **g** The specific interactions of DA2S in the 0 subsite highlighting the tryptophan cradle. In panels (**c**), (**d**), (**f**), and (**g**) residues specifically comprising the S-subsite are shown as violet sticks and those comprising the additional subsites are shown as blue (S1_19B) or green (S1_NC) sticks. Calcium ions are shown as a yellow sphere, and hydrogen bonds as dashed lines. In panels (**b**)–(**g**) sugar residues and subsites are labeled in green and red, respectively. **h** TLC analysis of ι-NC2 conversion by the total cellular fraction (TCF) from PS47. TCF* indicates heat inactivated TCF. κ-NC2 incubated with S1_19B, which produces β-NC2, is shown as a standard.

reducing end disaccharide (Supplementary Fig. 10d). The ι-NC2 represented in the model spanned the catalytic machinery while its poise in the active site revealed complete sequestration of the non-reducing end DA2S residue in a deep pocket, showing an obligate exo-mode of activity on carrageenan chain ends (Fig. 4e). The 2-sulfate of this DA2S unit in the 0 subsite was positioned relative to the S subsite amino acid sidechains in a manner anticipated for catalysis (Fig. 4f). The DA2S unit in the 0 subsite is complemented by the shape of the active site pocket while excluding solvent and is the only component of the ι-NC2 substrate that makes interactions with residues in the active site pocket of S1_NC (Fig. 4g). Though we could model the G4S residue of the disaccharide, S1_NC makes no direct or water-mediated interactions with this sugar residue, or with the G4S sulfate ester substituent (Fig. 4g), indicating a lack of plus ( + ) subsites and that the minimum recognition requirement for S1_NC is a single non-reducing end DA2S residue.

Though we were unable to produce active recombinant S1_NC, given this knowledge of its probable substrate, we tested PS47 cell extracts for carrageenan sulfatase activity. We observed conversion of ι-NC2 to a product having a TLC mobility consistent with β-neocarrabiose (β-NC2), most likely indicating desulfation of both monosaccharide residues by the sequential activities of S1_NC and S1_19B (Fig. 4h and Supplementary Fig. 6b). Taken together, these results support assignment of S1_NC as an exo-DA2S carrageenan sulfatase that would likely have activity on any form of carrageenan having a non-reducing end DA2S residue (e.g., ι- or α-carrageenan).

**Analysis of a β-neocarrabiose releasing exo-carrageenase**. A conserved feature of the CarPUL is a gene encoding a putative protein that has distant amino acid sequence similarity (<20% amino acid sequence identity) to GH42 β-galactosidases and 63% amino acid sequence identity to a carrageenan-active enzyme from *P. hydrolytica* S66[T] that is referred to as "GH42-like"[36]. These proteins, represented by locus tag EU509_08810 in PS47, constitute a newly classified GH family in the Carbohydrate-Active Enzymes Database[38], GH167, which in keeping with similarity to GH42 enzymes belongs to clan GH-A. Recombinant PS47 EU509_08810, referred to as GH167, appeared to be

and C84S mutants and attempted to determine structures of this protein in complex with neocarrageenan oligosaccharides and fragments thereof [Supplementary Notes (section 4) and Supplementary Fig. 10a–c]. This ultimately led to the structural analysis of the S1_NC C84S mutant in complex with ι-NC4.

The complex of S1_NC C84 with ι-NC4 revealed electron density for the substrate that enabled modeling of the non-

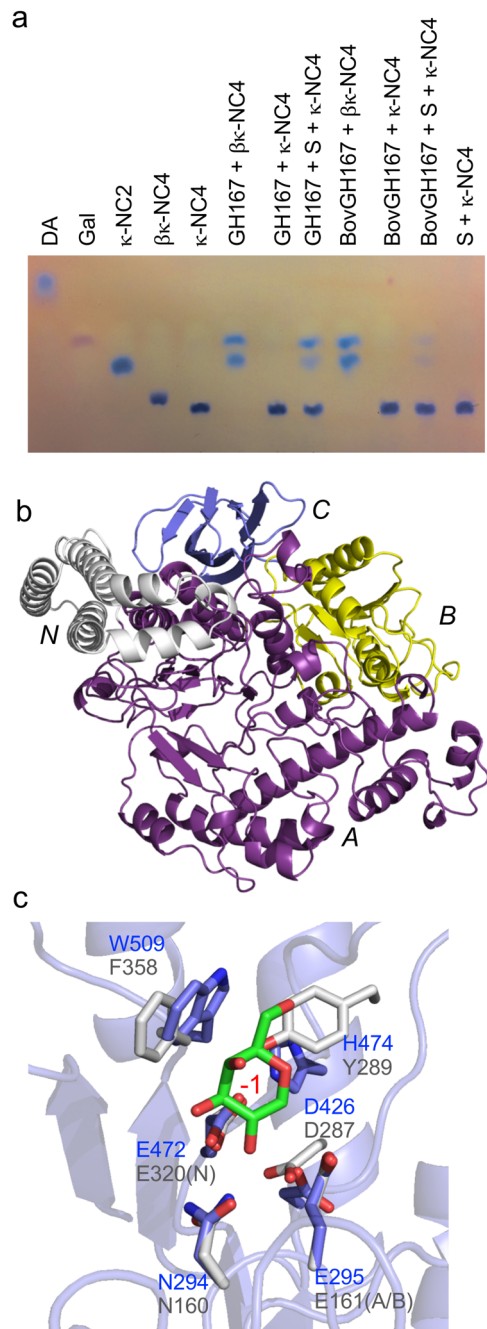

**Fig. 5 Activity and structure of β-neocarrabiose releasing exo-β-galactosidases. a** TLC analysis of GH167 and BovGH167 activity. "S" indicates the addition of S1_19B to the reactions. **b** Cartoon representation of the BovGH167 structure showing the domain organization and secondary structure elements. The N-terminal domain (domain N) is shown in gray, the (α/β)₈ domain (domain A) in purple, the mixed α/β-fold domain (domain B) in yellow, and the C-terminal β-sandwich domain (domain C) in blue. **c** Overlap of the BovGH167 active site (blue) with BbgII from *Bifidobacterium bifidum* S17 in complex with galactose in the −1 subsite (gray; PDB ID 4UCF). The catalytic acid/base (A/B) and nucleophile (N) for BbgII are indicated.

GH167 resulted in products with mobilities consistent with those of κ-NC2 and, therefore, likely β-NC2 (Fig. 5a). Further supporting this, GH167 was inactive on κ-NC4 but after treatment of κ-NC4 with S1_19B, which removes the 4-sulfate from the internal G4S residue converting the tetrasaccharide to β/κ-NC4, GH167 again produced products consistent with the formation of κ-NC2 and β-NC2 (Fig. 5a). The activity of GH167 thus appears to have the same activity as the *P. hydrolytica* "GH42-like" enzyme whereby through hydrolysis of the β-1,4-linkage, the enzymes release β-NC2 from the non-reducing end of β-carrageen and/or hybrid carrageenans[36].

We were unable to crystallize GH167 from PS47 so to investigate the molecular determinants for activity on β/κ-NC4 we utilized a homolog from *Bacteroides ovatus* CL02T12C04 (BovGH167), which originates from the same putative carrageenan PUL as the BovGH16 carrageenase, possesses 36% amino acid sequence identity with GH167, including a conserved putative active site (Supplementary Fig. 11a–c), and has the same enzymatic activity as GH167 (Fig. 5a). The structure of BovGH167 revealed four domains, three of which are often seen within the GH42 family (Fig. 5b). The conserved domains are domain A, the catalytic (α/β)₈-barrel domain; domain B, a 5-stranded parallel β-sheet surrounded by α-helices; and domain C, an anti-parallel β-sandwich. The unique domain is an all α-helical region at the N-terminus. Structural alignment of BovGH167 with BbgII from *Bifidobacterium bifidum* S17 (PDB ID 4UCF[39]) revealed that E295 and E472 [E288 and E473 in GH167 (numbering for the predicted matured protein)] overlap with the acid/base and the nucleophile, respectively, with additional conserved residues in the −1 subsite (Fig. 5c). Generation of glutamine substitution mutants at the putative nucleophile in GH167 and BovGH167 resulted in inactive proteins (Supplementary Fig. 11d), confirming the key importance of these residues to catalysis and the location of the conserved −1 subsite. Furthermore, and consistent with the membership of GH167 in clan GH-A, it points to a retaining catalytic mechanism that is shared with family 42 GHs.

**Characterization of a 3,6-anhydro-D-galactose dehydrogenase, DauA.** The first step in processing the free DA monosaccharide that would result from complete depolymerization of carrageenan is performed by a 3,6-anhydro-D-galactose dehydrogenase, which is putatively encoded by the *dauA* gene (locus tag EU509_08920 in PS47) in the CarPUL of *Pseudoalteromonas* sp.[33]. Recombinant DauA displayed activity on DA when using NADP⁺ as a co-factor (Supplementary Notes and Supplementary Fig. 12). DauA crystallized as a dimer with each monomer of the dimer having a deep NADP⁺ binding cleft [Fig. 6a and Supplementary Notes (section 5)]. The co-factor binding cleft leads to a tunnel that transits the core of the protein and opens to its opposite surface (Fig. 6b). The binding of NADP⁺ brings the nicotinamide ring in proximity to a series of residues that based on similarity to other aldehyde dehydrogenases comprise the catalytic machinery[40] (Fig. 6b and Supplementary Fig. 13a, b). A series of residues lining this catalytic pocket, which are also conserved within *Z. galactanivorans* DauA, are likely candidates for binding and positioning DA for catalysis (Fig. 6b and Supplementary Fig. 13c).

## Discussion

The presence of a genetic locus having an architecture consistent with carrageenan metabolism is quite widespread amongst marine *Pseudoalteromonas* species[33]. However, the model carrageenan degrading pseudoalteromonad, *P. carrageenovora* 9ᵀ, which degrades carrageenan, appears to lack the ability to utilize κ- or ι-carrageenan as a sole carbon source. Thus, our primary question

inactive against κ- and ι-carrageenan, even when pre- or co-treated with S1_19B, as well as being inactive on the synthetic β-galactosidase substrate pNP-β-galactopyranoside. Using TLC we found activity on hybrid β/κ-neocarratetraose (β/κ-NC4), which has a non-reducing end β-NC2 motif (Fig. 5a). Treatment with

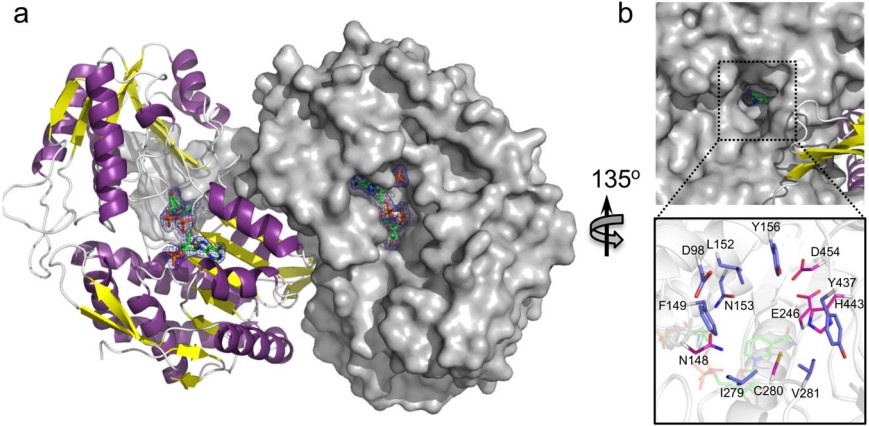

**Fig. 6 Structure of DauA. a** The dimer of DauA in complex with NADP$^+$ shown with one monomer in cartoon representation and the second monomer as its solvent accessible surface. α-helixes are colored purple, β-sheets colored yellow, and loops are colored gray. The NADP$^+$ to each monomer is shown in stick representation with its corresponding electron density shown as a $2F_o-F_c$ map contoured at 1σ (blue mesh). **b** Close-up of the active site where 3,6-anhydro-D-galactose binding occurs. Catalytic residues are shown as magenta sticks and residues likely involved in 3,6-anhydro-D-galactose binding as blue sticks.

was whether the presence of a CarPUL is generally linked to carrageenan depolymerization and whether under other circumstances it confers the ability to yield energy from the polysaccharide. Therefore, we isolated five new *Pseudoalteromonas* strains (the fifth, PS2, likely a duplicate), with all but one isolate possessing the CarPUL, and dissected their capacity to utilize carrageenan. We further focused on the molecular details underpinning the κ/ι-carrageenan metabolism pathway in one strain, *P. fuliginea* PS47.

*P. fuliginea* PS42 failed to grow under any condition when provided with κ- or ι-carrageenan as the carbon source, consistent with its natural deficiency of the CarPUL. The four strains having the CarPUL, PS47, and PS2 (*P. fuliginea* subspecies), U2A (*P. distincta* subspecies), and FUC4 (indeterminate species of *Pseudoalteromonas*), displayed growth on both ι- and κ-carrageenan, which supports the concept that these strains have fully functional κ/ι-carrageenan metabolism pathways that enable the bacteria to liberate energy from this algal polysaccharide. However, the growth phenotypes of the CarPUL-containing strains were complex. These strains grew on ι-carrageenan in solution, but only when provided with small amounts of κ-carrageenan oligosaccharide (but not ι-carrageenan oligosaccharide) or with in situ depolymerization of the ι-carrageenan by exogenous addition of a carrageenase (BovGH16) to the culture. Notably, no strain grew when provided only with ι- or κ-carrageenan oligosaccharides. This suggests that low levels of κ-carrageenan oligosaccharides can regulate the metabolism of highly polymerized ι-carrageenan, though how this may occur is unknown (Fig. 7a). In situ enzymatic depolymerization of the ι-carrageenan by BovGH16 promoted growth; however, we believe it likely that this promiscuous carrageenase also releases κ-carrageenan oligosaccharides from κ-carrageenan contaminants present in the ι-carrageenan.

In contrast to growth on ι-carrageenan, our CarPUL-containing *Pseudoalteromonas* strains grew on gelled κ-carrageenan without the addition of exogenous oligosaccharides. The deployment of extracellular enzymes for carrageenan hydrolysis (GH16A and/or GH16C), which show properties of κ-carrageenan specificity, may contribute to the regulation of carrageenan metabolism through the generation of oligosaccharides (Fig. 7a). However, we note that none of the genes encoding full-length GH16 enzymes in U2A showed significant transcript levels in the absence of carrageenan, making it unclear if "surveillance levels" of an extracellular κ-carrageenase is produced to aid in the initial response of the

bacterium to the presence of carrageenan. Alternatively, our *Pseudoalteromonas* strains may respond directly to highly polymerized κ-carrageenan or the solid κ-carrageenan medium may already contain enough free κ-carrageenan oligosaccharide fragments to regulate carrageenan metabolism.

Though our *Pseudoalteromonas* strains grow on polymeric ι- and κ-carrageenan, they do not grow on exogenously provided ι- or κ-carrageenan oligosaccharides, which points to an inability to import oligosaccharides (at least those with a DP of 8 or less). Furthermore, though these pseudoalteromonads can utilize ι-carrageenan, there is no evidence that our strains of *Pseudoalteromonas* have a mechanism to depolymerize ι-carrageenan in the extracellular environment. Together, these observations support the concept that growth on ι- and κ-carrageenan relies on the outer membrane transport system to import highly polymerized carrageenan into the periplasm to be utilized (Fig. 7a). The extracellular deployment of κ-carrageenase(s), in addition to potentially generating regulatory κ-carrageenan oligosaccharides, may aid in generating a higher abundance of polysaccharide chain ends that can thread through the transporter. If and how this may occur for ι-carrageenan in our *Pseudoalteromonas* strains is unclear, but it may simply rely on already available chain ends. Overall, our interpretation is that our *Pseudoaltermonas* strains primarily to respond to and utilize κ-carrageenan with the consumption of ι-carrageenan being a secondary capacity. Furthermore, it implicates what has been referred to as a "selfish" mechanism of polysaccharide metabolism because it minimizes potential sharing of polysaccharide breakdown products amongst the wider bacterial community[41].

In the periplasm κ-carrageenan is likely directly depolymerized by periplasmic GH16A and/or GH16C. However, both ι- and κ-carrageenan are substrates for the S1_19A endo-G4S ι-carrageenan sulfatase[28]. Through desulfation of either class of carrageenan by S1_19A, the polysaccharide becomes a substrate for GH16B, which is predicted to be a lipid-anchored protein oriented into the periplasmic space and displays properties consistent with activity on α- and β-carrageenan (or likely also hybrid carrageenans, such as furcellaran) (Fig. 7a). The use of a sulfatase to alter the class of ι-carrageenan prior to depolymerization by a specific endo-carrageenase is a distinct solution to dealing with this polysaccharide. In *P. carrageenovora* 9$^T$ the GH82 enzyme is postulated to assist with ι-carrageenan depolymerization[33].

The combined action of S1_19A and the periplasmic GH16 enzymes would generate a pool of oligosaccharides, likely with

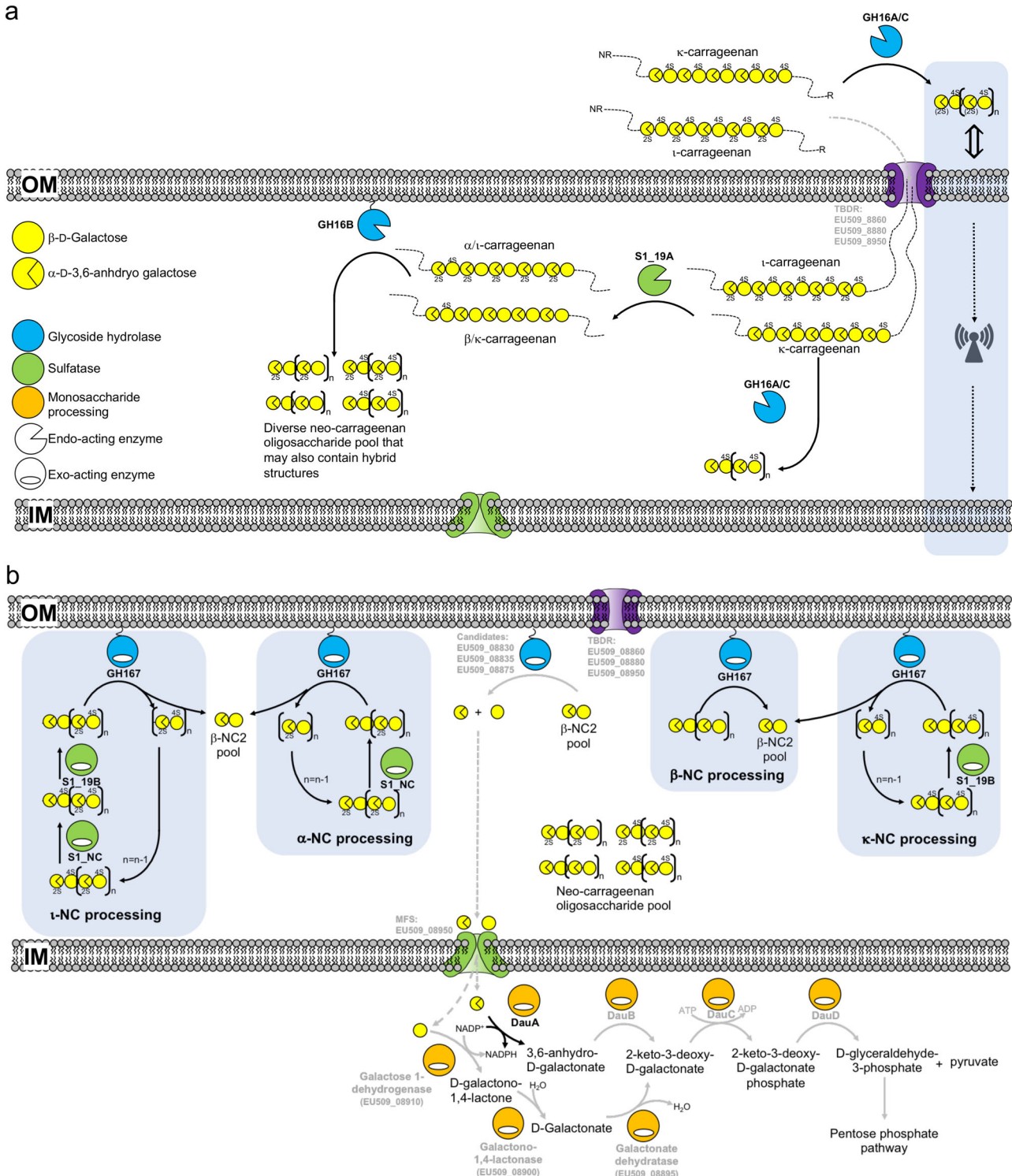

**Fig. 7 Model of carrageenan metabolism by *Pseudoalteromonas fuliginea* PS47 and other *Pseudoalteromonas* sp. that possess a CarPUL. a** Model of carrageenan depolymerization steps. A signal for the induction of carrageenan metabolism is provided by κ-carrageenan or a κ-carrageenan oligosaccharide through an unknown mechanism (indicated by the blue shaded area on the right of the panel), though this likely involves the TonB-dependent receptor complex. Polymeric carrageenan is imported and subsequently depolymerized to a heterogeneous pool of neocarrageenan oligosaccharides. **b** Model of sequential carrageenan oligosaccharide desulfation and depolymerization steps. Steps indicated by gray arrows and gray text are those that are not presently supported by experimental evidence.

quite varied sulfation patterns that potentially range from fully sulfated (ι-carrageenan oligosaccharides) to various hybrid sulfations to a complete lack of sulfation (β-carrageenan oligosaccharides). The S1_19B and S1_NC exo-sulfatases likely work

individually, or sequentially, on the termini of carrageenan oligosaccharides to generate a non-reducing terminal β-NC2 motif (Fig. 7b). Though we postulate that this is the most likely route to generate this motif, we also note that S1_19A, while primarily

notable for its endo-activity, also has activity on ι-neocarrageenan oligosaccharides and can remove the 4-sulfate from the G4S residue that is adjacent to the non-reducing end of ι-NC4 [see Supplementary Notes (section 6)]. This would generate an α-neocarrabiose motif at the non-reducing end that S1_NC would likely desulfate to a non-reducing terminal β-NC2 motif, thus providing a potential alternate route involving S1_19A. Oligosaccharides with this motif at the non-reducing terminus would then be hydrolyzed by GH167, which is predicted to be membrane associated via a lipid anchor, to release the non-reducing terminal β-NC2 motif. Continued sequential sulfatase and GH167 activity would completely reduce the oligosaccharides to a pool of β-NC2 and free sulfate (Fig. 7b).

Reduction of the β-NC2 pool to monosaccharides through the action of a α-1,3-(3,6-anhydro)-D-galactosidase is a necessary step in the liberation of energy from ι- and κ-carrageenan. At present, enzymes having this activity are classified into GH127 and GH129[32]; the lack of homologs in the *P. carrageenovora* 9[T] CarPUL was hypothesized to be the cause of this bacterium's inability to grow on carrageenan[33]. The lack of GH127 and GH129 homologs appears to be a general property of the *Pseudoalteromonas* CarPUL; however, the ability of our isolates to grow on carrageenan argues that α-1,3-(3,6-anhydro)-D-galactosidase activity must be present. In PS47, we hypothesize that the most likely candidates for this activity are EU509_08830, EU509_08835, and/or EU509_08875. These are conserved amongst all of the *Pseudoalteromonas* CarPULs, two of them have upregulated expression on carrageenan (the third has significant transcripts present in the absence of carrageenan), they are predicted to have 6- or 7-bladed β-propeller folds that are adopted by other GH families, and they are predicted to be lipid-anchored periplasmic proteins [see Supplementary Notes (section 7)]. Unfortunately, we have been unable to generate soluble and active recombinant versions of these proteins, nor have we been able to detect conclusive α-1,3-(3,6-anhydro)-D-galactosidase activity in PS47 total cellular fractions. Thus, the enzyme(s) associated with this activity remain(s) to be uncovered in *Pseudoaltermonas* sp. However, the PS47 CarPUL does encode an active NADP[+] dependent 3,6-anhydro-D-galactose dehydrogenase (DauA) that would initiate cytoplasmic processing of free DA. Additionally, as noted by Gobet et al.[33], the CarPUL encodes all of the enzymes for DA processing, and separate enzymes for galactose processing through the De Ley-Douderoff pathway, with these two pathways likely converging at 2-keto-3-deoxy-D-galactonate (Fig. 7b)[42]. All of these monosaccharide processing enzymes are upregulated in U2A upon growth on ι-carrageenan, further supporting the concept of a fully functional metabolic pathway.

Biochemical mapping of the majority of the steps in this pathway reveals key differences between this pathway and that of *Z. galactanivorans*. Specifically, pseudoalteromonads deploy a β-NC2 releasing β-carrageenase instead of an exo-β-D-galactosidase, use a different complement of carrageenan-specific sulfatases, and have an exo-α-1,3-(3,6-anhydro)-D-galactosidase from a different family (though it remains to be identified in species of *Pseudoalteromonas*) [see Supplementary Notes (section 8) for expanded details]. The *Pseudoalteromonas* species described here also have a unique solution for depolymerizing ι-carrageenan, which involves a sulfatase-catalyzed class switch of carrageenan type, though the lack of an ι-carrageenan-specific endo-hydrolase is not a conserved feature of all *Pseudoalteromonas* CarPULs. Overall, the evidence supports the existence of a fully functional CarPUL that imparts upon four new *Pseudoalteromonas* isolates the ability to utilize κ- and ι-carrageenan as an energy source, which is in keeping with the capacity of this genus to colonize macroalgal surfaces in the environment.

## Methods

**Materials**. κ-neocarrabiose, κ-neocarratetraose, β/κ-neocarratetraose, κ-neocarrahexaose, κ–ι–κ-neocarrahexaose, and κ-neocarraoctaose were obtained from V-Labs (Covinton, LA). κ-carratriose was obtained from Qingdao BZ Oligo Biotech Co (Qingdao, China). ι-neocarrabiose and ι-neocarratetraose were generated as described previously[28]. All reagents, chemicals and other carbohydrates were purchased from Sigma unless otherwise specified.

**Isolation, genome sequencing, and annotation**. *P. fuliginea* PS47 was isolated and its genome sequenced and annotated as described previously[28,43]. Isolation and genome sequencing of *P. fuliginea* PS2 was performed using the same procedures; Illumina-based sequencing and genome assembly was performed by the Genome Sciences Centre of the British Columbia Cancer Agency as described previously. *P. distincta* U2A and *Pseudoalteromonas* sp. FUC4 were also isolated using procedures previously described[43]. Genomic DNA was extracted from *P. distinctica* U2A and *Pseudoalteromonas* sp. FUC4 grown in 5 mL Zobell Marine Broth, incubated for 20 h at 25 °C, 200 rpm, using the DNeasy Blood & Tissue Kit (Qiagen, Germany). Paired-end DNA libraries were prepared using NEBNext® dsDNA Fragmentase (New England BioLabs, USA), SPRIselect® Reagent Kit (Beckman Coulter, Inc. USA) and NEBNext® Ultra[TM] II DNA Library Prep Kit for Illumina® (New England BioLabs, USA). Library quantification was done using the Qubit[TM] dsDNA HS assay kit (Invitrogen, USA), the NEBNext Library Quant Kit (New England BioLabs, USA) and an Agilent 2100 Bioanalyzer High Sensitivity DNA chip (Agilent Technologies, Inc., USA). The sequencing of libraries was performed on the Illumina MiSeq platform with the MiSeq V2 Reagent Kit (500-cycles) (Illumina Inc., USA). Sequencing analysis of paired-end reads and genome assembly was performed using the A5-miseq pipeline[44] or SPAdes[45].

For all isolates, the list of DNA scaffolds was submitted to the RAST server for identification of open reading frames and automated annotation. dbCAN (database for Carbohydrate-active enzyme ANnotation)[46] was then used for more detailed annotation of the putative carbohydrate-active enzymes (CAZyme). Putative CAZymes and protein products of co-localized genes in the carrageenan loci were further annotated using BLASTp and PHYRE2[47].

Species identification was performed by 16S RNA-based comparison using EzBioCloud[48]. For each isolate, the whole-genome average nucleotide identity with the top six 16S RNA hits were calculated to inform species identity[49].

**Growth of *Pseudoalteromonas* isolates on carrageenan**. Growth of *Pseudoalteromonas* isolates on ι-carrageenan and carrageenan oligosaccharides was assessed at 25 °C in a SpectraMax M5 plate reader at 600 nm essentially as previously described[50]. Quadruplicate overnight cultures of each strain were grown in Zobell broth from individual colonies, washed with minimal marine medium (MMM)[51] and resuspended in MMM containing no carbon source. Dialyzed and lyophilized ι-carrageenan (0.4% w/v) and oligosaccharides (0.4% w/v) were prepared in MMM and filter sterilized prior to use. Washed cells were used to inoculate 100 μL cultures 1/50 in 96-well microplates, and sealed plates were incubated for 60 h with shaking and OD readings every 20 min. Control wells contained uninoculated carrageenan/oligosaccharide, carbohydrate-free MMM inoculated with bacteria, and MMM containing 0.4% (w/v) galactose inoculated with bacteria. When necessary, filter-sterilized BovGH16 was added to wells at a final concentration of 2 μM. Growth of strains on κ-carrageenan was assessed on solid medium. κ-carrageenan (1% w/v), with or without 0.5% (w/v) galactose, was prepared in MMM and briefly boiled in a microwave to dissolve prior to pouring into petri dishes. Plate inocula were prepared by growing strains in MMM containing either 0.5% (w/v) galactose or iota carrageenan +0.04% (w/v) κ-NC4 for 60 h, washing cells in MMM containing no carbon source, and preparing 1/10 serial dilutions in carbon-free MMM. Spots (8 μL) were dropped onto dried plates, allowed to soak in and then incubated at 25 °C for 7 days.

**RNA sequencing**. For RNAseq experiments, pilot cultures set-up as described above were monitored for density using a test tube spectrophotometer in order to determine the OD corresponding to the mid-log phase of growth. For RNA extraction, duplicate cultures of U2A originating from different colonies were grown in 5 mL cultures in 16 mm test tubes in MMM containing either 0.2% (w/v) galactose, or 0.2% (w/v) ι-carrageenan plus BovGH16 (2 μM). Cultures were inoculated with 1/50 washed cells and incubated at 25 °C and 200 rpm until they reached mid-log, at which point cells were pelleted and flash frozen in liquid nitrogen.

RNA was extracted from two cell pellets for each treatment using the TRIzol Max Bacterial RNA Isolation Kit (Life Technologies, Carlsbad, CA, USA), according to the manufacturer's instructions. RNA extract quality and yield were assessed on a NanoDrop One/OneC Microvolume UV-Vis Spectrophotometer (Life Technologies, Carlsbad, CA, USA), and yields confirmed using the Qubit RNA HS Assay Kit and Qubit 2.0 fluorometer (Life Technologies, Carlsbad, CA, USA). Duplicate RNA extracts from each treatment were pooled to maximize yields prior to further processing. Residual DNA was removed from RNA extracts using the TURBO DNA-free kit (Life Technologies, Carlsbad, CA, USA), following the manufacturer's protocol for DNase treatment and subsequent DNase inactivation and removal.

mRNA enrichment was performed using the MICROBExpress Bacterial mRNA Enrichment Kit (Life Technologies, Carlsbad, CA, USA), and confirmed using the RNA 6000 Pico Kit on a Bioanalyzer 2100 Instrument (Agilent, Santa Clara, CA, USA). RNAseq libraries were generated using the Ion Total RNA-Seq v2 Kit (Life Technologies, Carlsbad, CA, USA) according to the manufacturer's instructions. Template preparation and loading onto an Ion 530 sequencing chip was done with an Ion Chef (Life Technologies, Carlsbad, CA, USA) using an Ion 510 & Ion 520 & Ion 530 Kit-Chef (Life Technologies, Carlsbad, CA, USA). Sequencing was performed on an Ion S5 System (Life Technologies, Carlsbad, CA, USA); demultiplexing and read trimming was performed with Torrent Suite 5.10.1. A total of 2.3–8.4 million single-end reads with an average read length of 129 bases yielded when 222 million and 1000 million Q20 bases per sample.

Transcriptome assembly was performed on the *P. distincta* U2A reference genome in Geneious Prime 2019.1.3 set to medium-low sensitivity with five fine tuning iterations. Reads with multiple matches were mapped randomly and gaps were allowed at 10% maximum per read (other parameters: 25 base minimum overlap, 18 base word length, 20% maximum mismatch per read, 15 base maximum gap size, 80% minimum overlap identity, 13 base index word length, 4 base maximum ambiguity). Expression levels were calculated in Geneious Prime 2019.1.3 by comparing transcripts normalized by the median of gene expression ratios as described in Dillies et al.[52]. Transcripts per million (TPM) were calculated according to Wagner et al.[53] as: TPM = (CDS read count * mean read length * $10^6$)/ (CDS length * total transcript count). $Log_2$-genome coverage files and number of reads mapping to each predicted CDS were calculated and differential expression analysis (normalization and statistical tests) was performed using Geneious. Genes with differential expression $|Log_2|$ ratios >1.5 with a $p$-value < 0.05 were considered differentially expressed.

**Detection of carrageenases in *Pseudoalteromonas* sp. culture.** *Pseudoalteromonas* strains were grown in 3 mL cultures of MMM containing 0.2% (w/v) ι-carrageenan plus 0.04% (w/v) κ-NC4 for 60 h as described above. Cells were pelleted at 13,000 rpm for 3 min, the supernatants retained and the pellets lysed in 150 μL BugBuster® Protein Extraction Reagent (Novagen) at room temperature for 20 min. To remove contaminating carbohydrate originating from the BugBuster®, lysed cell pellets were dialyzed against binding buffer (20 mM Tris pH 8.0, 500 mM NaCl) overnight at 4 °C in 100 μL dialysis buttons (Hampton Research) fitted with a 3 kDa MWCO membrane. Carrageenan and oligosaccharide digests contained 10 μL culture supernatant or lysed cell pellet in 20–50 μL reactions. Samples were analyzed by FACE or TLC as described below.

**Cloning and mutagenesis.** The gene encoding *Bacteroides ovatus* GH16 (BovGH16; accession number HMPREF1069_02099) was cloned as described previously[28]. Cloning of the gene encoding S1_19A into pET28a, production of the protein, and its purification was described previously[28]. The gene fragments lacking predicted signal peptides and encoding S1_19B (amino acids 28–473) and S1_NC (amino acids 34–507 were cloned using identical PCR-based procedures (See Supplementary Table 3 for primer sequences) as for S1_19A.

The gene fragment encoding for GH167, without the predicted signal peptide (amino acids 21–808), was amplified from PS47 genomic DNA and cloned into pET28a using the PIPE (Polymerase Incomplete Primer Extension) method[54] and the primers GHnew_fwd, GHnew_rev, Vector_primer1, and Vector_primer2 (Supplementary Table 3). The gene encoding *B. ovatus* GH167 (BovGH167; accession number HMPREF1069_02044) was synthesized as an *E. coli* codon optimized gene (GeneScript). The fragment encoding amino acids 26–775 was PCR amplified with BovGHnew_fwd and BovGHnew_rev primers (Supplementary Table 3) and cloned into pET28a via NheI and XhoI restriction sites.

The full gene encoding for DauA was amplified from PS47 genomic DNA (Supplementary Table 3) and cloned into pET28a by standard molecular biology procedures using NheI and XhoI restriction sites for directional cloning.

The gene fragments encoding for GH16A, GH16B, and GH16C, without the predicted signal peptides (amino acids 26–296, 34–346, and 26–397, respectively), were amplified from PS47 genomic DNA (see Supplementary Table 3 for primer sequences). These were cloned via the Quick In-Fusion Cloning Protocol (Takara-Clonetech, USA) into a custom pET28a maltose binding-protein (MBP) fusion construct. Specific primers for each construct were used to amplify the target vector (Supplementary Table 3). The final constructs encoded, from N- to C-terminus, an N-terminal MBP, a seven amino acid P/T linker, a TEV protease cleavage site, the GH16 module, and a 6-histidine tag.

All DNA amplifications were done using Phusion High-Fidelity DNA polymerase or CloneAmp HiFi PCR Premix. Mutations were created by site-directed mutagenesis (QuikChange Site-Directed Mutagenesis kit) (Supplementary Table 3). All constructs were confirmed for fidelity by bi-directional Sanger sequencing.

**Protein expression and purification.** Protein expressions for S1_19A, S1_19B, and S1_NC were performed using procedures identical to those previously described for S1_19A in the presence or absence of a co-produced recombinant formylglycine generating enzyme[28]. Briefly, expression plasmids were co-transformed into *E. coli* BL21 (DE3) Star with the plasmid pBAD/myc-his A Rv0712 (FGE) (Addgene plasmid #16132), which encodes a formylglycine

generating enzyme to promote sulfatase maturation. Transformed bacteria were grown in LB broth containing 50 μg mL$^{-1}$ kanamycin sulfate, 100 μg mL$^{-1}$ ampicillin, and 50 μg mL$^{-1}$ chloramphenicol at 37 °C with agitation at 180 rpm until the cell density reached an OD600 of ~0.5. The growth temperature was then dropped to 16 °C and FGE expression was induced with 0.02% L-arabinose. After ~2 h, sulfatase expression was induced with a final concentration of 0.5 mM iso-propyl-1-thio-β-D-galactopyranoside (IPTG) and the culture incubated for a further 16 h. For all other proteins, *E. coli* BL21 (DE3) Star transformed with plasmid were grown in 2 L cultures of LB broth containing 50 μg mL$^{-1}$ kanamycin sulfate at 37 °C with agitation at 180 rpm until cell density reached an OD$_{600}$ of ~0.5 at which time the temperature was dropped to 16 °C. After ~2 h, recombinant protein production was induced with a final concentration of 0.5 mM isopropyl-1-thio-β-D-galactopyranoside (IPTG) and the culture incubated for a further 16 h.

For all protein expressions, cultures were centrifuged at 8000 × g for 10 min. Harvested cells were chemically lysed by resuspension in 35% (w/v) sucrose, 1% (w/v) deoxycholate, 1% (v/v) Triton X-100, 500 mM NaCl, 10 mg lysozyme, and 0.2 μg mL$^{-1}$ DNase in 20 mM Tris (pH 8.0). The lysate was clarified by centrifugation at 16,000 × g for 30 min.

All S1 proteins, GH167 proteins, DauA and BovGH16 were purified by immobilized metal affinity chromatography. The clarified lysate was applied to a nickel affinity chromatography column and eluted with a step gradient of imidazole concentrations in 0.5 M NaCl, 20 mM Tris (pH 8.0). Eluted fractions were analyzed by sodium dodecyl sulfate polyacrylamide gel electrophoresis (SDS-PAGE), and samples containing the protein of interest were concentrated with a stirred ultrafiltration cell (EMD Millipore) fitted with a 10 kDa MWCO membrane. Further purification was performed using a HiPrep 16/60 Sephacryl S-200 HR size exclusion chromatography column (GE Healthcare) equilibrated with 0.5 M NaCl, 20 mM Tris (pH 8.0). Pure samples, as assessed by SDS-PAGE, and free of soluble aggregate were pooled and concentrated for use. N-terminal His-Tag was cleaved from all proteins used for crystallization by overnight incubation with thrombin in thrombin cleavage buffer [500 mM NaCl, 20 mM Tris-HCl (pH 8.0), and 2.5 mM CaCl$_2$] followed by size exclusion chromatography.

The GH16 MBP fusions were purified by amylose affinity chromatography. The clarified lysate was applied to amylose resin and washed with a solution comprising 20 mM Tris (pH 7.5), 250 mM NaCl, 1 mM EDTA, and 20% glycerol. Bound protein was eluted with wash buffer containing 10 mM maltose. Elution fractions were analyzed by SDS-PAGE, and samples containing the protein of interest were concentrated and buffer exchanged in a stirred ultrafiltration cell (EMD Millipore) fitted with a 10 kDa MWCO membrane.

**Enzyme assays.** Following incubation of carrageenan or oligosaccharides with cellular fractions or enzymes, fluorophore-assisted carbohydrate electrophoresis (FACE) was performed using methods as described[55]. Similarly, for thin layer chromatography (TLC) analysis samples were spotted on a silica gel plate and allowed to air dry. The silica gel plate was then placed in a chamber containing running buffer of formic acid, butanol, and water at a ratio of 8:4:1 or 2:2:1 acetic acid:butanol:dH$_2$O. The silica gel plate was dried and visualized using napthor-esorcinol with heating at 110 °C for 15 min.

Kinetic analysis of S1_19B activity on κ-NC2 was performed as described previously in a buffer containing 0.5 M NaCl, 1 mM 3-(N-morpholino) propanesulfonic acid (MOPS) (pH 7.16)[28]. Briefly, the release of protons from sulfate ester hydrolysis was spectrophotometrically monitored using para-nitrophenol (pNP) included in the MOPS buffer as an indicator. Dissolved carbonate was removed from the dH$_2$O used to prepare assay buffers by boiling for 30 min followed by cooling at 4 °C under nitrogen. Reactions contained 0.5 mM NaCl, 1.0 mM MOPS (pH 7.16), 0.8 mM pNP (500 mM stock prepared in DMSO), 0.01–2.5 mM κ-NC2, and 1.0 μM sulfatase. Initial rates were determined by monitoring a decrease in absorbance at 405 nm at 25 °C over 1 h. Standard curves were prepared by substituting carrageenan substrates with titrated amounts of HCl up to 1 mM. A linear relationship between proton produced and absorbance allowed rate determination with a correction factor applied account for buffer and indicator concentrations. Kinetic parameters were determined by nonlinear fitting of the Michaelis-Menten equation to the data. Values and errors represent the means and standard deviations, respectively, of experiments performed in triplicate.

**DauA activity and kinetics.** The activity of DauA was initially qualitatively screened using DA and several concentrations of NADP$^+$ or NAD$^+$. The formation of reduced co-factor was followed at 340 nm using a SpectraMax5 plate reader (Molecular Devices). Significant activity was only observed for NADP$^+$. Subsequently, the pH optimum for DauA activity was determined using 20 nM of enzyme, 400 μM of DA and 3 mM of NADP+ in McIlvaine's buffer (0.2 M disodium phosphate, 0.1 M citric acid) over a range of pH from 4.4 to 8.0[56], as well as in Tris buffer (20 mM) supplemented with 0.5 M NaCl over a pH range from 7.0 to 9.0.

Reaction mixtures for the determination of NADP$^+$ kinetic constants were set up in triplicate at 25 °C in 20 mM Tris pH 8.0 with 0.5 M NaCl in 100 μL volumes containing 20 nM of enzyme, 10 mM of 3,6-anhydro-D-galactose and 0–1000 μM of NADP +. Reaction mixtures for the determination of DA kinetic constants were set up in triplicate at 25 °C in 20 mM Tris pH 8.0 with 0.5 M NaCl in 100 μL volumes containing 20 nM of enzyme, 3 mM of NADP+, and 0–1000 μM of DA.

The formation of the reaction product was followed at 340 nm using a SpectraMax5 plate reader (Molecular Devices). The rate of release was determined by linear regression using the linear part of the curve. The extinction coefficient for NADPH used was 6476.2 $M^{-1}$ $cm^{-1}$. Michaelis-Menten parameters were determined by nonlinear curve fitting.

**Crystallization, diffraction data collection, and processing.** All crystals were grown at 18 °C by hanging drop vapor-diffusion in conditions outlined in Supplementary Table 4 with 1:1 ratios of crystallization solution and protein. Complexes of S1_19B were obtained by co-crystallization with an excess of carbohydrate, complexes of S1_NC were obtained by soaking crystals in crystallization solution supplemented with an excess of carbohydrate, and the complex of DauA was obtained by soaking crystals in crystallization solution supplemented with 2 mM $NADP^+$. Crystals were cryoprotected by soaking in the respective crystallization solution containing 25–32% ethylene glycol prior to flash-cooling in liquid nitrogen at 100 K for diffraction data collection. The BovGH167 iodide derivative was obtained by incorporating 1 M NaI into the cryoprotection solution.

Diffraction data were collected at the Canadian Light Source (CLS, Saskatoon, Saskatchewan) on beamline 08ID-1 and the Stanford Synchrotron Radiation Lightsource (SSRL, Stanford, California), on beamlines BL9-2 and BL11-1, as indicated in Supplementary Table 5. Alternately, diffraction data were collected on an instrument comprising a Pilatus 200 K 2D detector coupled to a MicroMax-007HF X-ray generator with a VariMaxTM-HF ArcSec Confocal Optical System and an Oxford Cryostream 800, which is indicated in Supplementary Table 5 as "In-house". All diffraction data collected at the CLS were processed using MOSFLM and SCALA[57]. Data collected at the SSRL and in-house were integrated, scaled and merged using HKL2000[58].

**Structure solution and refinement.** The structures of S1_19B and S1_NC were solved by molecular replacement using the program PHASER[59] and the coordinates of *P. fuliginea* PS47 S1_19A (PDB ID code: 6BIA)[28] as a search model. BUCCANEER[60] was used to automatically build models, which were finished by manual building with COOT[61] and refinement with REFMAC[62]. The finished models were then used to solve the structures of the mutants in complex with ligands by molecular replacement using the program PHASER.

The structure of BovGH167 was determined by the single isomorphous replacement with anomalous signal method using the native and iodide derivative data sets. Heavy atom substructure determination, refinement, phasing, density modification, and initial model building was performed with AutoSol within the PHENIX package[63]. The model resulting from this was finished by manual building with COOT and refinement with REFMAC using the higher resolution native data set.

The DauA structure was solved by molecular replacement using PHASER and the crystal structure of *E. coli* succinic semialdehyde dehydrogenase as a search model (PDB code 3JZ4)[64]. Automatic model building and model completion was performed as above except that refinements were performed using phenix.refine within the PHENIX package[65]. The structure of uncomplexed DauA was used as a molecular replacement model to determine the structure in complex with $NADP^+$.

For all structures, the addition of water molecules was performed in COOT with FINDWATERS and manually checked after refinement. In all data sets, refinement procedures were monitored by flagging 5% of all observations as "free"[66]. Model validation was performed with MOLPROBITY[67]. All data processing and model refinement statistics are shown in Supplementary Table 5.

**Statistics and reproducibility.** Growth experiments were done with $n = 4$ independent experiments and error was calculated as the standard error of the mean. Enzyme kinetics experiments were done with $n = 3$ independent experiments and error was calculated as the standard deviation. RNAseq experiments were performed with duplicate biological replicates; extensive statistical analyses were performed with Geneious Prime 2019.1.3 as described in Dillies et al.[52].

**Reporting summary.** Further information on research design is available in the Nature Research Reporting Summary linked to this article.

## Data availability

The coordinates and structure factors for all X-ray crystal structures have been deposited under the PDB IDs 6PNU, 6POP, 6PRM, 6PSM, 6PSO, 6PT4, 6PT6, 6PT9, 6PTK, and 6PTM. The *P. fuliginea* PS2, PS42, and PS47 Whole Genome Shotgun projects have been deposited in DDBJ/ENA/GenBank under the accessions SEUL00000000, SEUK00000000, and SEUJ00000000, respectively. The *P. distincta* U2A and *Pseudoalteromonas* sp. FUC4 Whole Genome Shotgun projects have been deposited under the accessions SEUH00000000 and SEUI00000000. RNA sequencing data has been deposited in SRA database under BioProject accession PRJNA576066.

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

## Acknowledgements

We thank the Genome Sciences Centre of the BC Cancer Agency for performing genome sequencing reactions and genome assembly. This research was supported by a Natural Sciences and Engineering Research Council of Canada Discovery Grant (FRN 04355). We thank the staff at the Canadian Light Source (CLS) and the Stanford Synchrotron Radiation Lightsource (SSRL; SLAC National Accelerator Laboratory) where diffraction data were collected. The CLS is supported by the Natural Sciences and Engineering Research Council of Canada, the National Research Council Canada, the Canadian Institutes of Health Research, the Province of Saskatchewan, Western Economic Diversification Canada, and the University of Saskatchewan. Use of the SSRL is supported by the U.S. Department of Energy, Office of Science, Office of Basic Energy Sciences under Contract No. DE-AC02-76SF00515. The SSRL Structural Molecular Biology Program is supported by the DOE Office of Biological and Environmental Research, and by the National Institutes of Health, National Institute of General Medical Sciences (P41GM103393). The contents of this publication are solely the responsibility of the authors and do not necessarily represent the official views of NIGMS or NIH.

## Author contributions

A.B.B. and A.G.H. designed and conceived the study, interpreted data, and wrote the paper. K.T.A., O.S.-A., J-H.H., C.V., J.K.H., A.G.H., E.B., J.V.H., and A.B.B. performed microbe isolation, genome sequencing, genome assembly, and genome annotation. C.V., J.K.H., E.B., and J.V.H. performed and analyzed RNA sequencing experiments. J.K.H. performed bacterial growth experiments. J.P.M.H., F.B., A.B., and J.Z. designed, performed and analyzed mass spectrometry experiments. A.G.H., B.E.M, and B.P. performed gene cloning and enzymology experiments. A.G.H. and B.P. performed structural studies. All authors had full access to the data and approved the paper before it was submitted by the corresponding author.

## Competing interests

The authors declare no competing interests.
