## [Peer Review File · Communications Biology]

Reviewers' comments:

Reviewer #1 (Remarks to the Author):

Summary

The authors have used various complementary approaches to study the kappa-/iota-carrageenan metabolic pathway in some *Pseudoalteromonas* that were isolated from red algae. They could show that 4 out of the 5 *Pseudoalteromonas* isolates present the necessary kappa-/iota-carrageenan-specific polysaccharide utilization locus CarPUL genes involved in the kappa-/iota-carrageenan pathway. These 4 isolates could grow on kappa-/iota-carrageenan when supplemented with an exogenous GH16 or some carrageenan oligosaccharides. They could show that growth on carrageenan upregulates the CarPUL and they could verify the activity of several enzymes involved of the pathway as well as specific biochemical and structural features of these enzymes.

General comments

First, I would like to apologize I wasn't able to comment on the structural features, the crystallography of the enzymes and the mass spectrometry, as these parts are quite far from my expertise.

Overall this is a comprehensive and interesting study which completes the current literature on kappa-/iota-carrageenan metabolism in marine bacteria.

One general comment is that the authors should not generalize their findings to the whole *Pseudoalteromonas* genus, starting from the title of the manuscript for instance. As they wrote in this manuscript and also in the literature they cite, the CarPUL and the ability to degrade kappa-/iota-carrageenan is restricted to some *Pseudoalteromonas* or maybe one clade. Also, the authors may not generalize the title about the carrageenan metabolism and they should specify that they investigated the kappa-/iota-carrageenan pathway. I am stressing this because it is different from the pathway of lambda-carrageenan, another polysaccharide commonly found in red macroalgal cell wall. For instance, they could attenuate the title this way: "The kappa-/iota-carrageenan metabolism pathway of some marine *Pseudoalteromonas* species". The authors should be more specific about these two points in the whole manuscript and correct it accordingly.

This is very minor but it would have been nice to have a line numbering in the text, it would have made it easier to comment on it.

Specific comments

Introduction

3rd paragraph: There is a typing mistake "in the saccharolytic bacteroidete*s* *Zobellia*".

4th paragraph: There is a typing mistake "with the ability of these *pseudoalteromonads* to liberate energy".

Results

1st paragraph: Following my comment above, the authors may modify "Marine *Pseudoalteromonas* species possess conserved carrageenan PULs." to "*Some* marine *Pseudoalteromonas* species possess conserved carrageenan PULs.". In the 3rd paragraph too: "*Some* marine *Pseudoalteromonas* species can grow using carrageenan as a sole carbon source." Also in the latter sentence of the 3rd paragraph:, the authors may remove "sole", I am not sure if this can be considered entirely true as the isolates are not able to grow on kappa-/iota-carrageenan only but they need either the addition of an exogenous carageenase or some supplement with somre kappa-/iota-carrageenan oligosaccharides.

Last sentence of the 3rd paragraph: This sentence may be rephrased because iota-carrageenan is not the sole carbon source in the medium if there is a supplement of kappa-NC4 or kappa-NC8.

10th paragraph: This sentence is a bit difficult to follow, it may be rephrased: "Subsequently, we determined the structure of S1_19B in complex with intact kappa-NC2 substrate through the use of a mutant that was inactivated by a serine substitution of residue C77, which would otherwise be matured to the catalytic formylglycine (FGly) residue."

Discussion

1st paragraph: Please rephrase: "However, the model carrageenan degrading pseudoalteromonad, *P. carrageenovora* 9T, which is known to degrade carrageenan, appears to lack the ability to utilize kappa- or iota-carrageenan as a *SOLE* carbon source."

Methods

5th paragraph: The first sentence lacks the reference: "plate reader at 600 nm essentially as previously described (pectin paper ref) »

Figures & tables

Figure 2: In the legend, I am not sure what are the differences in culture conditions between panels (b) and (d), please rephrase or be more specific.

Figure 7: Very nice and comprehensive figure of the different mechanisms in the kappa-/iota-carrageenan pathway but some of the polices may be a bit enlarged. I am afraid it might not be readable once the paper will be printed.

Suppl. Table 1: the text in some cells is not entirely readable.

Reviewer #2 (Remarks to the Author):

The major claims of the paper are that "the metabolic systems that confer the ability to metabolize ... photosynthetically fixed carbon are not yet fully understood" and that isolates that possess a specific k/i-carrageenan PUL can grow on carrageenan.

The findings in this study are partly novel. As described below, several papers describe the degradation of carrageenan. However, this manuscript contains some new observations (e.g. on growth of bacteria on carrageenan and structure of enzymes).

The work seems to be solid and convincing, and although it is not ground breaking new, several novel findings that would be of interest to others in the community are reported.

Peter Stougaard
Department of Environmental Science
Aarhus University
Denmark

The authors use a broad variety of methods to describe how the marine polysaccharide carrageenan is metabolized by four *Pseudoalteromonas* isolates. The authors focus on the first part in the degradation pathway from polymeric kappa- and iota-carrageenan to dimeric beta-neocarrabiose. Classical microbiological growth experiments with different poly- and oligomeric carrageenans were used in combination with genomics and functional and structural analyses of recombinant enzymes.

Degradation of carrageenan by marine bacteria has been described before. Several papers describe single components in the degradation pathway, and at least four papers describe whole or parts of the carrageenan degradation pathway. Lee et al (2016; doi: 10.1007/s00253-016-7346-6) as the first described the pathway from 3,6-anhydro-D-galactose (DA), Ficko-Blean et al (2017; doi: 10.1038/s41467-017-01832-6) described carrageenan degradation in *Zobellia galactanivorans*, Gobet et al. (2018; doi: 10.3389/fmicb.2018.02740) in *Pseudoalteromonas carrageenovora* 9T, and Schultz-Johansen et al. (2018; doi: 10.3389/fmicb.2018.00839) in *Paraglaciecola hydrolytica* S66T.

What is new in this manuscript is

- Growth of pseudoalteromonads in iota-carrageenan supplemented with neocarraoligosaccharides.
- Crystals and 3D structures of two sulfatases and a 3,6 anhydro-D-galactose dehydrogenase.
- Activity studies of sulfatases and a 3,6-anhydro-D-galactose dehydrogenase.

The manuscript focuses on the first part of the carrageenan degradation pathway and the enzymes involved in degradation and desulfatation of kappa- and iota-carrageenan to beta neocarrabiose. Furthermore, the step from 3,6-anhydro-D-galactose to 3,6-anhydro-D-galactonate catalyzed by a 3,6-anhydro-D-galactose dehydrogenase (*dauA*) was investigated. Description of all the subsequent catabolic steps including hydrolysis of beta-neocarrabiose is based on published data and hypotheses put forward in this manuscript.

General, major comments:

1) Genome sequences and structure data are not publically available. Thus, I have not had a chance of evaluating sequences and sequence comparisons or molecular structures.

2) Re. Growth of *Pseudoalteromonads* on carrageenans. In this study, it is shown that four of the isolates can grow with iota-carrageenan as carbon source if supplemented with kappaneooligosaccharides (k-NC4 and k-NC8). Gobet et al. 2018 showed that *P. carrageenovora* 9T, which contains a similar CarPUL, did not grow on kappa- or iota-carrageenan, and they concluded that *P. carrageenovora* 9T was "unable to use kappafamily carrageenans alone". However, Supplementary Table 2 in this manuscript shows that the CarPULs in *P. carrageenovora* 9T and in the four pseudoalteromonads from this study contain the same putative 3,6-anhydro-D-galactosidase genes. Thus, *P. carrageenovora* 9T and the four pseudoalteromonads from this study most probably would show the same growth pattern if they were cultivated with kappa- or iota-carrageenan as carbon sources if supplemented with kappaneooligosaccharides (k-NC4 and k-NC8). I agree that the data presented in the manuscript are sufficient to conclude that the four pseudoalteromonads are able to utilize and to grow on iota-carrageenan. However, it would strengthen the conclusions if the four isolates were cultivated in kappa-carrageenan with/without kNC4 and k-NC8 and if *P. carrageenovora* 9T was included in the study. If the authors have such data, they should be included in the manuscript.

3) Re. The missing 3,6-anhydro-D-galactosidase. This study and the paper by Gobet et al 2018 failed to identify an enzymatic activity responsible for the degradation of beta-neocarrabiose. As mentioned above, Gobet et al concluded that *P. carrageenovora* 9T is unable to use kappa-carrageenan as carbon source because of the missing ability to grown on kappa-carrageenan and the lacking 3,6-anhydro-Dgalactose hydrolase gene. However, this study argues that since the four pseudoalteromonads can grow in iota-carrageenan, gene(s) that encode degradation of beta-neocarrabiose must be present in the four isolates. Three putative genes are hypothesized to be candidates, EU509_8830, EU509_8835 and EU509_8875. The argument is that the proteins may contain a lipid-anchor that directs the protein to the periplasm. However, no documentation of localization of the proteins is presented. This is very speculative and it would have been nice if additional arguments supported the hypothesis

that one or more of the proteins could be the new beta-neocarrabiase activity. For example, did the RNAseq data on isolate 47 support the fact that the three genes were upregulated when cultivated in iota-carrageenan? This is very difficult to see from the RNAseq data in Supplementary Table 3, since the numbering seems to be different from the text (EU509_88XX in text vs. EU511_XXXX in Supplementary Table 3).

Are the two EU-numbering (in text and in Supplementary Table 3) different or identical?

The linkage to be hydrolyzed in beta-neocarrabiase is an alpha(1-3) linkage and not a beta(1-4) as in all the other enzyme activities described (kappa-, iota, beta/kappa-carrageenases). Analyses (Blast, HHPred, etc.) of the putative structure of the proteins EU509_8830, EU509_8835 and EU509_8875 might give an idea of the activities when compared to related sequences and structures. Please include such analyses and use the results in the Discussion on enzyme activity and degradation pathway.

4) Re. Enzymes responsible for the subsequent degradation of 3,6-anhydro-D-galactose. The authors investigate the first step in the degradation of 3,6-anhydro-D-galactose (DA). Recombinant enzyme (DauA) was produced and analyzed with respect to activity and structure. They refer to the paper by Gobet et al 2018 for the subsequent steps. However, Gobet et al did not either document the degradation of DA but they referred to previous paper by Ficko-Blean et al. (doi: 10.1038/s41467-017-01832-6).

I believe that the original reference for the degradation of DA should be Lee et al. 2016 (doi: 10.1007/s00253-016-7346-6). Please add this reference.

Fig. 1 indicates that all four pseudoalteromonads and *P. carrageenovora* 9T contain the complete genetic repertoire for degradation of DA. If so, this should be mentioned in the text (e.g. page 18) and could be used as an argument for the fact that the pseudoalteromonads may contain the entire pathway for degradation of carrageenans despite the lacking 3,6-anhydro-D-galactosidase.

Furthermore, if the four dau genes were upregulated in the transcriptomics study, this could also be mentioned as an argument, and should be presented in the Results and Discussion sections.

5) Re. Activity of GH16B enzyme. It is shown (Fig. 3) that the GH16B enzyme (similar to the "furcellaranases" described by Schultz-Johansen et al. 2018) is active with kappa- and iota-carrageenan treated with an S1_19A sulfatase but not on pure kappa- and iota-carrageenan. The sulfatase is claimed to be G4S-specific but no documentation is presented for this activity. Also, the sulfatase may not be 100% efficient and could result in partially desulfated kappa-carrageenan (beta/kappa hybrid, furcellaran-like) or iota-carrageenan (alpha/iota hybrid). Thus, the substrates for the GH16B enzyme in Fig. 3 may be hybrid and not totally desulfated polysaccharides.

Do you have documentation (NMR analyses?) of the structure of the sulfatase treated substrates? If so, please include in the results section.

It is hypothesized that the difference in migration of the products seen in reactions where the substrates are co-treated vs. pre-treated with sulfatase is due to "post-depolymerization desulfatation" (Fig. 3c), indicating that the products are G4S sulfated. This is seen for the kappacarrageenases (GH18A and GH18C), but in reactions with the GH16B enzyme this is not very obvious. In Fig.3d this difference is not seen indicating that the products released by the GH18B enzyme do not contain a G4S sulfate. Please explain in the text what is happening and what you think

is the product.

NMR analyses of the products would be nice to document the specific activity of the GH18B enzyme. If such data exist, please include in the result section.

6) Re. GH42-like enzyme. The activity of the PS47 GH42-like enzyme was shown to be similar to that of the *Paraglaciecola hydrolytica* GH42-like enzyme. However, the PS47 enzyme could not be crystallized and therefore the authors analyzed the structure of a similar enzyme, BovGH42L, from *Bacteroides ovatus*. However, *Pseudoalteromonas* and *Bacteroides* inhabit very different habitats, algal surfaces/marine environments vs. human intestines, and thus they have different nutrient requirements. The two enzymes display only 36% identity and they show different activities: Figure 5 shows that the PS47 enzyme hydrolyze k-NC4 treated with an S1_19B sulfatase (lane 8) whereas the Bov42L enzyme does not (lane 11).

Therefore, the two enzymes may not be very similar and it is questionable how much information on the BovGH42L enzyme is transferable to the PS47 GH42L enzyme. Unless additional enzymatic investigations can document that the two enzymes display identical enzymatic activities, it should be mentioned in the Discussion that the two enzymes may not be similar, or alternatively the structure data should be taken out and presented in a more relevant context.

Minor comments:

The authors denote the enzymes "carrageenases". When searching databases, "carrageenase" returns 1 (PubMed) and 848 (Google) hits, whereas "carrageenanase" results in 86 (PubMed) and ca. 16,600 (Google) hits. This may be a small linguistic issue, but the authors should consider to rephrase to "carrageenanase" throughout the manuscript in order to improve the searchability.

In the Results section "Growth on carrageenan upregulate the CarPUL" part, the authors speculate whether there is "surveillance" levels of pathway components. This part should be re-written and the more speculative parts removed to Discussion.

Figure 2 legend mentions that the concentration of galactose is 1%, but in Materials and Methods it is 0.5%. Which is correct? Font size in Supplementary Table 1 and 2 is very small. Please enlarge.

Reviewer #3 (Remarks to the Author):

"The Carrageenan Metabolism Pathway of Marine *Pseudoalteromonas* Species" by Hettle et al presents a multi-disciplinary characterization of many components of the carrageenan degrading machinery encoded in the *Pseudoalteromonas* genome. The authors have isolated 4 strains of this bacterium growing on macroalgae, analysed their genomes, transcriptomes and growth phenotypes on carrageenan and gone through to expressing individual enzymes present in a proposed PUL for structural and biochemical characterisation. Overall, I think this is a very well performed study and reaches the high bar that has been set by many exemplary pieces of work that have emerged over the past few years dissecting the biochemistry of diverse PULs. The manuscript is also very well written and clear which only left me with a few queries that I have listed below. I hope that if these can be addressed, it might help further draw out some of the key points of the paper before publication in *Communications Biology*.

INTRODUCTION

When discussing previous work in *Zobellia* in the sentence "The genes in this PUL and enzymes

deployed by it include endo acting carrageenases, exo-acting carrageenan-specific GHs and carrageenan specific sulfatases that confer upon the microbe the ability to depolymerize and metabolize κ - and ι -carrageenan." Perhaps it would be useful to include the GH and sulfatase classifications that present the activities discussed in parentheses or somehow to make it clear that there are additional GHs in this pull that are distinct from the others discussed previously and those discussed in this work? Otherwise, the first time the reader meets GH127 and 129 in this paper is in the discussion.

RESULTS

In the section "Marine Pseudoalteromonas species possess conserved carrageenan PULs" there is no comparison of the Pseudoalteromonas PUL to the Zobelia PUL which as far as I understand it represents the best characterized CarPUL so far. Could a comparison be included here to make it clearer where the distinctions are between the gene contents of the PULs?

The sub-heading "Marine Pseudoalteromonas species can grow using carrageenan as a sole carbon source." In my opinion is slightly misleading as the bugs can only grow when supplemented with an exogenous GH16, carrageenan oligosaccharides or when the substrate is solid. This is clearly a complex phenotype which is difficult to capture in a sub-heading but perhaps something like "Marine Pseudoalteromonas species display a complex phenotype when using Carrageenan as a sole carbon source" would be more appropriate?

In the structural analysis of S1_NC the authors state that "Initial X-ray crystallographic analysis of S1_NC suggested this was due to aberrant maturation of the proto-catalytic cysteine (Supplementary Notes and Supplementary Figures 5 and 6). Despite extensive efforts, we were unable to circumvent generation of this inappropriately matured form of the protein." Could it please be made clearer whether this is a common problem for the expression of sulfatases even when co-expressing the formyl-glycine generating enzyme? And what exactly was done to try and improve the maturation of this enzyme? I think it would also be useful to include an accurate mass spectrum of the protein in the supplementary information if available? This would show whether the cysteinic acid observed in the crystal structure is present in the native protein as purified as this could also be the result of radiation damage during X-ray data collection. Either way the protein will be inactive but I think it would be useful for readers that might be having similar problems elsewhere to understand these issues.

In the section "Identification and structure of a β -neocarrabiose releasing exo-carrageenanase" the authors describe the protein as GH42-like with 20% sequence ID to β -galactosidases in this family. I wonder, is this distinct enough to form a new GH family and have the authors approached the CAZy team to look into this? Might be worth doing? If they're not in a new family then surely the proteins are GH42 members in which case there is no need for the "-like"?

DISCUSSION

Paragraphs 2 and 3 – the authors first state that the organisms can grow on the polysaccharides but then go on to qualify this statement by describing the complex phenotypes in the next paragraph. This could be a little confusing, I think it would be better to describe the phenotype more clearly first and then to discuss the effects of cell free extracts etc on carrageenan and how this might suggest that the bugs require highly polymerised substrate to be able to metabolise it.

In paragraph 4 the authors discuss an apparent lack of background expression of the GH16s which may provide surveillance levels of protein to help control gene expression. Could the authors say whether there is any evidence that the secreted hypothetical proteins encoded by the PUL might, or might not, play a role here? Or are there possible genes outside of the PUL that might hint at this role?

In paragraph 4 the authors discuss the apparent inability of the bacterium to take up short oligosaccharides hinting that perhaps the organism can only metabolise highly polymerised form of the polysaccharide. This seems highly unusual to me, are there any other examples of bacteria that display this sort of behaviour on any other carbon sources? I don't disagree with the interpretation of the data, it would just be interesting to know whether this has been observed anywhere else?

In paragraph 8 the authors quite rightly discuss the lack of GH127s and 129s and how this impacts the ability of the organism to utilise the anhydro-galactosidases. It's a shame that the authors were not able to express EU509_8830, EU509_8835, and/or EU509_8875 which they identify as candidate enzymes that could play this role. They state that these proteins are predicted to encode 5- or 6-bladed beta propeller proteins. These are very common folds across diverse enzymes and could indicate that they are GHs or could have diverse other functions. Could the authors please provide some more information on how these predictions were performed? Is there any significant predicted similarity to specific GH families which may lend further support to the role of these proteins in releasing the DA units from the substrate?

SUPPLEMENTARY INFO

There are extensive notes here that are useful to the reader. Could the authors please number the sub-sections and refer to those specific sections in the main text to make it easier for the reader to find what they are being directed to? I missed some of these sections when reading through the first time.

In supplementary table 3 the highlighting of the transcripts that are identified from the PUL in grey is not overly helpful. Can I suggest that the authors highlight these with the same colour scheme used in figure 1 (blue for GHs, etc) to make it easier for the reader to pick out the GHs, the sulfatases, etc. This will require a change in colour for the hypothetical proteins in figure 1 but will allow the reader to locate each gene of interest to them more easily.

Authors' responses are in italics.

General response: Thank you to the reviewers for the many excellent suggestions of additional discussion points. Unfortunately, we are limited in the length the manuscript can be, which makes for a significant challenge with a study that is rich with data and accompanying interpretation, such as this. However, we have tried very hard to incorporate the additional analysis and/or discussion as sections in the Supplementary Notes, as indicated in the individual points below.

Reviewer #1 (Remarks to the Author):

General comments

One general comment is that the authors should not generalize their findings to the whole *Pseudoalteromonas* genus, starting from the title of the manuscript for instance. As they wrote in this manuscript and also in the literature they cite, the CarPUL and the ability to degrade kappa-/iota-carrageenan is restricted to some *Pseudoalteromonas* or maybe one clade. Also, the authors may not generalize the title about the carrageenan metabolism and they should specify that they investigated the kappa-/iota-carrageenan pathway. I am stressing this because it is different from the pathway of lambda-carrageenan, another polysaccharide commonly found in red macroalgal cell wall. For instance, they could attenuate the title this way: "The kappa-/iota-carrageenan metabolism pathway of some marine *Pseudoalteromonas* species". The authors should be more specific about these two points in the whole manuscript and correct it accordingly. This is very minor but it would have been nice to have a line numbering in the text, it would have made it easier to comment on it.

Response: We have altered the title as suggested and amended the text where appropriate. We have also added in line numbers.

Specific comments

Introduction

3rd paragraph: There is a typing mistake "in the saccharolytic bacteroidete*s* *Zobellia*".

4th paragraph: There is a typing mistake "with the ability of these *pseudoalteromonads* to liberate energy".

Response: Thank you for the good catches – we have made these corrections.

Results

1st paragraph: Following my comment above, the authors may modify "Marine *Pseudoalteromonas* species possess conserved carrageenan PULs." to "**Some* marine *Pseudoalteromonas* species possess conserved carrageenan PULs.". In the 3rd paragraph too: "**Some* marine *Pseudoalteromonas* species can grow using carrageenan as a sole carbon source." Also in the latter sentence of the 3rd paragraph:, the authors may remove "sole", I am not sure if this can be considered entirely true as the isolates are not able to grow on kappa-/iota-carrageenan only but they need either the addition of an exogenous carageenase or some supplement with some kappa-/iota-carrageenan oligosaccharides.

*Response: In keeping with above, the section titles have been altered to "A carrageenan PUL that is conserved amongst some marine *Pseudoalteromonas* species." and "Some marine *Pseudoalteromonas* species display a complex growth phenotype when using κ - or ι -carrageenan as a carbon source."*

Last sentence of the 3rd paragraph: This sentence may be rephrased because iota-carrageenan is not the sole carbon source in the medium if there is a supplement of kappa-NC4 or kappa-NC8.

Response: Sole has been removed.

10th paragraph: This sentence is a bit difficult to follow, it may be rephrased: "Subsequently, we determined the structure of S1_19B in complex with intact kappa-NC2 substrate through the use of a mutant that was inactivated by a serine substitution of residue C77, which would otherwise be matured to the catalytic formylglycine (FGly) residue."

Response: Yes, this was a bit of a long sentence. We have broken it up to read: "Subsequently, we determined the structure of S1_19B in complex with intact κ -NC2 substrate. This was enabled by the use of a mutant that was inactivated by a serine substitution of residue C77, which would otherwise be matured to the catalytic formylglycine (FGly) residue."

DISCUSSION

1st paragraph: Please rephrase: "However, the model carrageenan degrading pseudoalteromonad, *P. carrageenovora* 9T, which is known to degrade carrageenan, appears to lack the ability to utilize kappa- or iota-carrageenan as a *SOLE* carbon source."

Response: Sole has been inserted.

METHODS

5th paragraph: The first sentence lacks the reference: "plate reader at 600 nm essentially as previously described (pectin paper ref) »

Response: Thank you for finding this – the reference has been inserted.

FIGURES & TABLES

Figure 2: In the legend, I am not sure what are the differences in culture conditions between panels (b) and (d), please rephrase or be more specific.

Response: The difference between panels b and d are that the medium in b also contained galactose. This was indicated in the figure legend but before the description of the two panels together. We have moved this distinction after the description of the two panels and feel it is now clearer: "In panels (b) and (d), prior to spotting on the plates, the bacteria were pre-grown in liquid medium comprising MMM, 0.4% ι -carrageenan and 0.04% κ -NC4. In panel (b) the solid medium was also supplemented with 1% galactose."

Figure 7: Very nice and comprehensive figure of the different mechanisms in the kappa-/iota-carrageenan pathway but some of the police may be a bit enlarged. I am afraid it might not be readable once the paper will be printed.

Response: We have tried to increase as many of the font sizes as possible, though this is challenging in a complex figure. We are partly relying on this journal being in digital "print" where font size is not as much of a problem.

Suppl. Table 1: the text in some cells is not entirely readable.

Response: We now fixed this.

Reviewer #2 (Remarks to the Author):

General, major comments:

1) Genome sequences and structure data are not publicly available. Thus, I have not had a chance of evaluating sequences and sequence comparisons or molecular structures.

Response: Apologies to the reviewer that this was felt as lacking. We have had serious and unfortunate problems in the past with early public release of our data where, between the time of manuscript submission and public release of the article, other groups have incorporated our results into their studies – unattributed – and rushed their articles to publication before ours. We are diligent about making sure our data is always publicly available, but do so during the manuscript revision phase. As such, we have authorized public release of the genome sequences and the structural data; they should now be available.

2) Re. Growth of Pseudoalteromonads on carrageenans. In this study, it is shown that four of the isolates can grow with iota-carrageenan as carbon source if supplemented with kappaneooligosaccharides (k-NC4 and k-NC8). Gobet et al. 2018 showed that *P. carrageenovora* 9T, which contains a similar CarPUL, did not grow on kappa- or iota-carrageenan, and they concluded that *P. carrageenovora* 9T was “unable to use kappafamily carrageenans alone”. However, Supplementary Table 2 in this manuscript shows that the CarPULs in *P. carrageenovora* 9T and in the four pseudoalteromonads from this study contain the same putative 3,6-anhydro-D-galactosidase genes. Thus, *P. carrageenovora* 9T and the four pseudoalteromonads from this study most probably would show the same growth pattern if they were cultivated with kappa- or iota-carrageenan as carbon sources if supplemented with kappa neooligosaccharides (k-NC4 and k-NC8). I agree that the data presented in the manuscript are sufficient to conclude that the four pseudoalteromonads are able to utilize and to grow on iota-carrageenan. However, it would strengthen the conclusions if the four isolates were cultivated in kappa-carrageenan with/without kNC4 and k-NC8 and if *P. carrageenovora* 9T was included in the study. If the authors have such data, they should be included in the manuscript.

*Response: We certainly agree with the reviewer that *P. carrageenovora* would most likely grow on carrageenan under the conditions we used. In truth, we intentionally avoided doing this experiment out of respect for the authors of the Gobet article – we did not want to be seen as trying to outright prove this group as wrong. Rather, we left this as something that group could follow up on, if they wish. We do believe, however, that our results are robust enough that readers will come to the same conclusion that the reviewer did.*

3) Re. The missing 3,6-anhydro-D-galactosidase. This study and the paper by Gobet et al 2018 failed to identify an enzymatic activity responsible for the degradation of beta-neocarrabiose. As mentioned above, Gobet et al concluded that *P. carrageenovora* 9T is unable to use kappa-carrageenan as carbon source because of the missing ability to grown on kappa-carrageenan and the lacking 3,6-anhydro-Dgalactose hydrolase gene. However, this study argues that since the four pseudoalteromonads can grow in iota-carrageenan, gene(s) that encode degradation of beta-neocarrabiose must be present in the four isolates. Three putative genes are hypothesized to be candidates, EU509_8830, EU509_8835 and EU509_8875. The argument is that the proteins may contain a lipid-anchor that directs the protein to the periplasm. However, no documentation of localization of the proteins is presented. This is very speculative and it would have been nice if additional arguments supported the hypothesis that one or more of the proteins could be the new beta-neocarrabiase activity. For example, did

theRNAseq data on isolate 47 support the fact that the three genes were upregulated when cultivated in iota-carrageenan? This is very difficult to see from the RNAseq data in Supplementary Table 3, since the numbering seems to be different from the text (EU509_88XX in text vs. EU511_XXXX in Supplementary Table 3).

Are the two EU-numbering (in text and in Supplementary Table 3) different or identical?

Response: Just to clarify for the reviewer, and as we state in the results section, RNAseq was performed only on Pseudoalteromonas distincta U2A. Generating libraries for transcriptomic analysis of bacteria grown on highly anionic polysaccharides proves to be somewhat of a challenge. This, combined with a weaker growth phenotype of Pseudoalteromonas fuliginea PS47, led to early failure of transcriptomic experiments with PS47. RNAseq did, however, work with U2A, likely because of a more robust growth phenotype that allowed harvesting of larger number of bacteria at an appropriate growth phase. Thus, all of the accession codes correspond to those of U2A, not PS47; we have double checked the accuracy.

On the suggestion of reviewer #3 we have now used more extensive color coding for function in figure 1 and supplementary tables 2 and 3. The putative α -1,3-(3,6-anhydro)-D-galactosidases are now identifiable in all of these display items by their cyan color. An examination of figure 1 and supplementary table 3 will quickly reveal that two of the putative α -1,3-(3,6-anhydro)-D-galactosidase encoding genes in U2A have upregulated expression on iota-carrageenan while the third has significant transcripts under all tested conditions.

The linkage to be hydrolyzed in beta-neocarrabiose is an alpha(1-3) linkage and not a beta(1-4) as in all the other enzyme activities described (kappa-, iota, beta/kappa-carrageenases). Analyses (Blast, HHPred, etc.) of the putative structure of the proteins EU509_8830, EU509_8835 and EU509_8875 might give an idea of the activities when compared to related sequences and structures. Please include such analyses and use the results in the Discussion on enzyme activity and degradation pathway.

Response: Our hypothesis in the discussion regarding the function of these putative proteins was indeed based a variety of predictive analyses, which are mainly encompassed in the comprehensive Phyre2 fold recognition pipeline (which bundles all of the reviewer's suggested methods and more). We have now provided a section regarding this in the Supplementary Notes (section 7), which supports the relevant portion of the discussion.

4) Re. Enzymes responsible for the subsequent degradation of 3,6-anhydro-D-galactose. The authors investigate the first step in the degradation of 3,6-anhydro-D-galactose (DA). Recombinant enzyme (DauA) was produced and analyzed with respect to activity and structure. They refer to the paper by Gobet et al 2018 for the subsequent steps. However, Gobet et al did not either document the degradation of DA but they referred to previous paper by Ficko-Blean et al. (doi: 10.1038/s41467-017-01832-6). I believe that the original reference for the degradation of DA should be Lee et al. 2016 (doi: 10.1007/s00253-016-7346-6). Please add this reference.

Response: Yes, there is a paper trail here and the reviewer is quite correct. We have inserted this reference.

Fig. 1 indicates that all four pseudoalteromonads and P. carrageenovora 9T contain the complete genetic repertoire for degradation of DA. If so, this should be mentioned in the text (e.g. page 18) and could be used as an argument for the fact that the pseudoalteromonads may contain the entire pathway for degradation of carrageenans despite the lacking 3,6-anhydro-D-galactosidase.

Furthermore, if the four dau genes were upregulated in the transcriptomics study, this could also be mentioned as an argument, and should be presented in the Results and Discussion sections.

Response: With respect to the reviewer, we believe this was the essence of what was communicated at the end of the first paragraph on page 18. To further clarify this, we have amended the end of this paragraph to read:

“However, the PS47 CarPUL does encode an active NADP⁺ dependent 3,6-anhydro-D-galactose dehydrogenase (DauA) that would initiate cytoplasmic processing of free DA. Additionally, as noted by Gobet et al., the CarPUL encodes all of the enzymes for DA processing and separate enzymes for galactose processing through the De Ley-Douderoff pathway, with these two pathways likely converging at 2-keto-3-deoxy-D-galactonate (Figure 7b). All of these monosaccharide processing enzymes are upregulated in U2A with growth on ι-carrageenan, further supporting the concept of a fully functional pathway.”

Hopefully this is more in-line with what the reviewer is thinking. This section is appropriately referenced in the main text, including the Lee et al. reference.

5) Re. Activity of GH16B enzyme. It is shown (Fig. 3) that the GH16B enzyme (similar to the “fucellaranases” described by Schultz-Johansen et al. 2018) is active with kappa- and iota-carrageenan treated with an S1_19A sulfatase but not on pure kappa- and iota-carrageenan. The sulfatase is claimed to be G4S-specific but no documentation is presented for this activity. Also, the sulfatase may not be 100% efficient and could result in partially desulfated kappa-carrageenan (beta/kappa hybrid, fucellaran-like) or iota-carrageenan (alpha/iota hybrid). Thus, the substrates for the GH16B enzyme in Fig. 3 may be hybrid and not totally desulfated polysaccharides.

Do you have documentation (NMR analyses?) of the structure of the sulfatase treated substrates? If so, please include in the results section.

Response: Apologies to the reviewer regarding the S1_19A sulfatase activity. S1_19A is actually very well biochemically and structurally characterized [see Hettle, A. G. et al. The molecular basis of polysaccharide sulfatase activity and a nomenclature for catalytic subsites in this class of enzyme. Structure 26 1–12 (2018)]. We cited this study later in the manuscript and I am not sure how we missed incorporating this citation at this most relevant section of the manuscript – it is now added. All of the relevant details that reveal the action of S1_19B as a G4S-sulfatase on kappa- and iota-carrageenan, as well as NMR analysis of product, is given in this article. In the present submission this is supplemented with additional mass spectrometry data of oligosaccharide activity (see Supplementary Notes, section 6). Regarding potential incomplete removal of the G4S residues we mention in this section of the results that GH16B may be active on b/k-carrageenan or a/i-carrageenan, thus acknowledging potential activity on hybrid polysaccharides. To clarify this, we have added the word “hybrid” after each use of b/k-carrageenan or a/i-carrageenan.

It is hypothesized that the difference in migration of the products seen in reactions where the substrates are co-treated vs. pre-treated with sulfatase is due to “post-depolymerization desulfatation” (Fig. 3c), indicating that the products are G4S sulfated. This is seen for the kappacarrageenases (GH18A and GH18C), but in reactions with the GH16B enzyme this is not very obvious. In Fig.3d this difference is not seen indicating that the products released by the GH18B enzyme do not contain a G4S sulfate. Please explain in the text what is happening and what you think is the product.

NMR analyses of the products would be nice to document the specific activity of the GH18B enzyme. If such data exist, please include in the result section.

Response: This is a good observation. We have added the following sentence to this section: "Unlike with GH16A and GH16C, the pretreatment of ι-carrageenan with S1_19A followed by GH16B digestion, or cotreatment with both enzymes at the same time, resulted in the same pattern of GH16B products. Because GH16B activity depends on S1_19A activity, we suggest that the former observation indicates that the products of GH16B lack G4S residues that are susceptible to S1_19A activity, and may even entirely lack G4S residues."

Unfortunately, the GH16B enzyme (all three GH16s actually) was exceedingly difficult to express in significant quantities, making it a challenge to perform more detailed enzymatic analyses. Thus, we do not have NMR data on the products.

6) Re. GH42-like enzyme. The activity of the PS47 GH42-like enzyme was shown to be similar to that of the *Paraglaciecola hydrolytica* GH42-like enzyme. However, the PS47 enzyme could not be crystallized and therefore the authors analyzed the structure of a similar enzyme, BovGH42L, from *Bacteroides ovatus*. However, *Pseudoalteromonas* and *Bacteroides* inhabit very different habitats, algal surfaces/marine environments vs. human intestines, and thus they have different nutrient requirements. The two enzymes display only 36% identity and they show different activities: Figure 5 shows that the PS47 enzyme hydrolyze k-NC4 treated with an S1_19B sulfatase (lane 8) whereas the Bov42L enzyme does not (lane 11).

Therefore, the two enzymes may not be very similar and it is questionable how much information on the BovGH42L enzyme is transferable to the PS47 GH42L enzyme. Unless additional enzymatic investigations can document that the two enzymes display identical enzymatic activities, it should be mentioned in the Discussion that the two enzymes may not be similar, or alternatively the structure data should be taken out and presented in a more relevant context.

Response: Apologies again to the reviewer. The issue of the B. ovatus enzyme activity appears to have resulted from poor reproduction of figure 5 when collating the manuscript for submission. We have ensured in this submission that the image has been reproduced properly. Also note that these enzymes will be assigned an official CAZy GH family number and, for now, are referred to as GHxxx.

*The reviewer will see in the improved figure 5a that BovGHxxx does process k-NC4 in an S1_19B sulfatase dependent manner with the same products as for GHxxx (see Figure 5a, lane 11). The amount of product is quite low but this mainly results from incomplete activity of S1_19B; the incomplete processing capacity of S1_19B is also evident in figure 4a. A similar result is seen with the PS47 enzyme, albeit with a bit more product. Also, the product profiles for the two enzymes are identical in all cases. Thus, the activities of the two enzymes are undeniably similar, and arguably identical. Furthermore, the active site residues identified in the BovGHxxx structure are conserved with the PS47 enzyme (we have now provided an alignment and additional details in the supplementary to support this – please see the new supplementary figure 8 panels a-c). Mutation of the predicted conserved nucleophile in both enzymes resulted in the same inactive phenotype, again supporting the similarities. Finally, BovGHxxx is present in a PUL in *B. ovatus* that contains a number of predicted carrageenan active enzymes, including the BovGH16 carrageenase that we used in this study and another published study (the Hettle et al article). While analysis of this complex *B. ovatus* PUL is beyond the scope of this study we have made a comment regarding these features in the results to clearly support the relationship between*

*BovGHxxx function and carrageenan metabolism. It now reads: “We were unable to crystallize GHxxx from PS47 so to investigate the molecular determinants for activity on β/κ -NC4 we utilized a homolog from *Bacteroides ovatus* CL02T12C04 (BovGHxxx), which originates from the same putative carrageenan PUL as the BovGH16 carrageenase, possesses 36% amino acid sequence identity with GHxxx, including a conserved putative active site (Supplementary Figure 8a-c), and has the same enzymatic activity as GHxxx (Figure 5a).” We have also extensively edited lines 369 to 393 for clarity and to incorporate the new classification of family GHxxx by the CAZy team.*

Finally, with respect to the issue of bacterial habitat, the capability, genetics, and biochemistry of algal polysaccharide metabolism by members of the human gut microbiome is widely and thoroughly documented in numerous quite high profile articles (for example, please see PMIDs 23150581, 22393053, 29535379, 21747801, 20376150, 29795267, 29743671, 30110640). In fact, PULs from these microbes, which were likely obtained by horizontal gene transfer from marine microbes, have made excellent model systems for understanding the microbial metabolism of algal polysaccharides. We feel this body of evidence strongly supports BovGHxxx as an appropriate model to aid in our initial understanding of the molecular details underpinning the activity of this new family of GHs, which is founded upon this class of carrageenan active enzyme.

Minor comments:

The authors denote the enzymes “carrageenanases”. When searching databases, “carrageenanase” returns 1 (PubMed) and 848 (Google) hits, whereas “carrageenase” results in 86 (PubMed) and ca.16,600 (Google) hits. This may be a small linguistic issue, but the authors should consider to rephrase to “carrageenase” throughout the manuscript in order to improve the searchability.

Response: Yes, there are strange inconsistencies (at least to us) in how these general classes of enzymes are referred to (agarose active enzymes are agarases, not agaroseases, but porphyran active enzymes are porphyranases, not porphyrases). In this case, the reviewer makes an excellent point regarding the precedent that has been set and we have changed carrageenanase to carrageenase. Thank you for this.

In the Results section “Growth on carrageenan upregulate the CarPUL” part, the authors speculate whether there is “surveillance” levels of pathway components. This part should be rewritten and the more speculative parts removed to Discussion.

Response: We have simply removed the term surveillance levels, as this is indeed speculation of function. The sentence following the observation of some transcripts in the absence of carrageenan but presence of galactose now reads: “This may result from some capacity of the monosaccharide to regulate components the PUL or that it reveals the constitutive production of some pathway components.” These are really the only two options to explain the observation of the transcripts.

Figure 2 legend mentions that the concentration of galactose is 1%, but in Materials and Methods it is 0.5%. Which is correct?

Response: Good catch – we have changed the figures legends to the correct value of 0.5%

Font size in Supplementary Table 1 and 2 is very small. Please enlarge.

Response: We have enlarged these tables.

Reviewer #3 (Remarks to the Author):

INTRODUCTION

When discussing previous work in *Zobellia* in the sentence “The genes in this PUL and enzymes deployed by it include endo acting carrageenases, exo-acting carrageenan-specific GHs and carrageenan specific sulfatases that confer upon the microbe the ability to depolymerize and metabolize κ - and ι -carrageenan.” Perhaps it would be useful to include the GH and sulfatase classifications that present the activities discussed in parentheses or somehow to make it clear that there are additional GHs in this pull that are distinct from the others discussed previously and those discussed in this work? Otherwise, the first time the reader meets GH127 and 129 in this paper is in the discussion.

Response: This is a good suggestion and we have now added the family classifications in parenthesis at this point in the introduction.

RESULTS

In the section “Marine *Pseudoalteromonas* species possess conserved carrageenan PULs” there is no comparison of the *Pseudoalteromonas* PUL to the *Zobellia* PUL which as far as I understand it represents the best characterized CarPUL so far. Could a comparison be included here to make it clearer where the distinctions are between the gene contents of the PULs?

*Response: This is a logical suggestion and one we considered in the original preparation of the manuscript. This did not bear fruit, however, as there is virtually no synteny between the *Z. galactanivorans* PUL and the CarPUL. Many of the carrageenan active *Z. galactanivorans* enzymes are encoded by genes that are scattered around the genome (in other gene clusters). Where there is a key cluster of genes (the focus of the Ficko-Blean article) the conservation between *Z. galactanivorans* and the pseudoalteromonads is limited. Given that this comparison was somewhat uninformative, and also that Gobet et al had already provided this comparison, we chose not to cover this ground again. The comparison that Gobet et al provided, however, is now referenced in the introduction.*

The sub-heading “Marine *Pseudoalteromonas* species can grow using carrageenan as a sole carbon source.” In my opinion is slightly misleading as the bugs can only grow when supplemented with an exogenous GH16, carrageenan oligosaccharides or when the substrate is solid. This is clearly a complex phenotype which is difficult to capture in a sub-heading but perhaps something like “Marine *Pseudoalteromonas* species display a complex phenotype when using Carrageenan as a sole carbon source” would be more appropriate?

*Response: In accordance with this suggestion, and the similar one from reviewer 1, we have re-titled this section as: “Some marine *Pseudoalteromonas* species display a complex growth phenotype when using κ - or ι -carrageenan as a carbon source.” It is a bit long but we think it better captures the content of this results section.*

In the structural analysis of S1_NC the authors state that “Initial X-ray crystallographic analysis of S1_NC suggested this was due to aberrant maturation of the proto-catalytic cysteine (Supplementary Notes and Supplementary Figures 5 and 6). Despite extensive efforts, we were unable to circumvent generation of this inappropriately matured form of the protein.” Could it please be made clearer whether this is a common problem for the expression of sulfatases even when co-expressing the formyl-glycine generating enzyme? And what exactly was done to try and

improve the maturation of this enzyme? I think it would also be useful to include an accurate mass spectrum of the protein in the supplementary information if available? This would show whether the cysteinic acid observed in the crystal structure is present in the native protein as purified as this could also be the result of radiation damage during X-ray data collection. Either way the protein will be inactive but I think it would be useful for readers that might be having similar problems elsewhere to understand these issues.

Response: The reviewer raises a complex issue. We did pursue analysis of S1_NC maturation by mass-spectrometry. Unfortunately, despite the experiment being done by renowned protein mass-spectrometrists, the protein alone did not behave well enough to get an accurate intact mass spectrum and the peptide that was to carry the modification was not detected in peptide mapping experiments, so it remains unclear when the aberrant modification was introduced. Indeed, this highly oxidized sidechain species may have been generated anywhere from production of the protein to the diffraction data collection. However, as the reviewer has grasped, we ceased this exploration because – no matter the mechanism - we were faced with inactive enzyme. The generality of the problem is difficult to discuss in this manuscript, let alone comment on here. Informal discussions among the community who are studying carbohydrate-specific sulfatases indicate that maturation may be a relatively common problem (at least for organisms that are not part of the human microbiome). I say “may” because it is difficult to know with a new enzyme if it is inactive or if one simply doesn’t have the correct substrate. This problem leads to unpublished negative results and, therefore, nothing we can definitively discuss in this manuscript. In fact, this was one of our motivations for incorporating our strange result as it begins a published discussion of this issue and informs the potential use of structural approaches to uncover the specificity of sulfatases that are difficult, or impossible, to obtain in an active form.

In the section “Identification and structure of a β -neocarrabiose releasing exo-carrageenanase” the authors describe the protein as GH42-like with 20% sequence ID to β -galactosidases in this family. I wonder, is this distinct enough to form a new GH family and have the authors approached the CAZy team to look into this? Might be worth doing? If they’re not in a new family then surely the proteins are GH42 members in which case there is no need for the “-like”?

*Response: This is a good point and an issue that we have contemplated. We went with GH42-like as this designation was used in a previous publication from another group who reported the properties of the *P. hydrolytica* homolog of our enzyme. However, in an effort to make sure we get this right, we have contacted the CAZy team and these enzymes will indeed be classified into a new GH family that belongs to clan GH-A. As per the CAZy team policy, at present it is assigned the designation GHxxx and the manuscript has been edited to reflect this. Also per CAZy policy, at the time of providing the final accepted manuscript it will be updated to indicate the assigned family number.*

DISCUSSION

Paragraphs 2 and 3 – the authors first state that the organisms can grow on the polysaccharides but then go on to qualify this statement by describing the complex phenotypes in the next paragraph. This could be a little confusing, I think it would be better to describe the phenotype more clearly first and then to discuss the effects of cell free extracts etc on carrageenan and how this might suggest that the bugs require highly polymerised substrate to be able to metabolise it.

Response: Upon careful consideration of this, we found the sentence regarding the cell free extracts entirely unnecessary. We have deleted this sentence and made rearrangements, which we feel improves the flow and clarity. Please see the top of page 15.

In paragraph 4 the authors discuss an apparent lack of background expression of the GH16s which may provide surveillance levels of protein to help control gene expression. Could the authors say whether there is any evidence that the secreted hypothetical proteins encoded by the PUL might, or might not, play a role here? Or are there possible genes outside of the PUL that might hint at this role?

Response: Unfortunately, we have no other convincing evidence regarding this. There are other CAZymes in these pseudoalteromonads, but nothing else obviously associated with carrageenan metabolism. There are possibilities, of course, but these would require extensive future investigations that follow on from this observation.

In paragraph 4 the authors discuss the apparent inability of the bacterium to take up short oligosaccharides hinting that perhaps the organism can only metabolise highly polymerised form of the polysaccharide. This seems highly unusual to me, are there any other examples of bacteria that display this sort of behaviour on any other carbon sources? I don't disagree with the interpretation of the data, it would just be interesting to know whether this has been observed anywhere else?

Response: We were also surprised and are unaware of such an observation being made previously.

In paragraph 8 the authors quite rightly discuss the lack of GH127s and 129s and how this impacts the ability of the organism to utilise the anhydro-galactosidases. It's a shame that the authors were not able to express EU509_8830, EU509_8835, and/or EU509_8875 which they identify as candidate enzymes that could play this role. They state that these proteins are predicted to encode 5- or 6-bladed beta propeller proteins. These are very common folds across diverse enzymes and could indicate that they are GHs or could have diverse other functions. Could the authors please provide some more information on how these predictions were performed? Is there any significant predicted similarity to specific GH families which may lend further support to the role of these proteins in releasing the DA units from the substrate?

Response: Reviewer #2 also made this request. The analysis is a bit lengthy so we have now added it to the supplementary material. We also corrected the typo where we wrote 5- and 6-bladed propellers – it should be 6- and 7-bladed propellers.

SUPPLEMENTARY INFO

There are extensive notes here that are useful to the reader. Could the authors please number the sub-sections and refer to those specific sections in the main text to make it easier for the reader to find what they are being directed to? I missed some of these sections when reading through the first time.

Response: This is a really good suggestion and we have incorporated it into this revision. Whether this complies with journal formatting policies, however, remains to be determined.

In supplementary table 3 the highlighting of the transcripts that are identified from the PUL in grey is not overly helpful. Can I suggest that the authors highlight these with the same colour scheme used in figure 1 (blue for GHs, etc) to make it easier for the reader to pick out the GHs, the sulfatases, etc. This will require a change in colour for the hypothetical proteins in figure 1 but will allow the reader to locate each gene of interest to them more easily.

Response: Another good idea that we have taken advantage of. Some genes have been recolored in Figure 1 and in the supplementary tables 2 and 3. We have also taken this opportunity to highlight the putative α -1,3-(3,6-anhydro)-D-galactosidases.

REVIEWERS' COMMENTS:

Reviewer #2 (Remarks to the Author):

The authors have responded acceptable to all the comments in my first review.

However, small adjustments still require attentions from the authors side: All sequences are publically available, but you have to correct some of the accession numbers, e.g. line 511: EU509_8830, EU509_8835, and EU509_8875, the numbers should be EU509_08830, EU509_08835, and EU509_08875. There may be other places in the manuscript, including the supplementary material, where the numbers should be corrected. Please check!

In conclusion, I think the work is well performed and the manuscript is well written.

Reviewer #3 (Remarks to the Author):

The authors have adequately addressed all the points that I raised so I recommend that this work be published.

Authors' responses are in italics.

Reviewer #2 (Remarks to the Author):

The authors have responded acceptable to all the comments in my first review.

However, small adjustments still require attentions from the authors side: All sequences are publically available, but you have to correct some of the accession numbers, e.g. line 511: EU509_8830, EU509_8835, and EU509_8875, the numbers should be EU509_08830, EU509_08835, and EU509_08875. There may be other places in the manuscript, including the supplementary material, where the numbers should be corrected. Please check!

In conclusion, I think the work is well performed and the manuscript is well written.

Response: Thank you to the reviewer for picking this up. We have made this correction throughout the manuscript.